# Checkpoint blockade and nanosonosensitizer-augmented noninvasive sonodynamic therapy combination reduces tumour growth and metastases in mice

Wenwen Yue[1,2], Liang Chen[3], Luodan Yu[2], Bangguo Zhou[1,2], Haohao Yin[1,2], Weiwei Ren[1], Chang Liu[1], Lehang Guo[1], Yifeng Zhang[1], Liping Sun[1], Kun Zhang [1], Huixiong Xu[1] & Yu Chen [2]

Combined checkpoint blockade (e.g., PD1/PD-L1) with traditional clinical therapies can be hampered by side effects and low tumour-therapeutic outcome, hindering broad clinical translation. Here we report a combined tumour-therapeutic modality based on integrating nanosonosensitizers-augmented noninvasive sonodynamic therapy (SDT) with checkpoint-blockade immunotherapy. All components of the nanosonosensitizers (HMME/R837@Lip) are clinically approved, wherein liposomes act as carriers to co-encapsulate sonosensitizers (hematoporphyrin monomethyl ether (HMME)) and immune adjuvant (imiquimod (R837)). Using multiple tumour models, we demonstrate that combining nanosonosensitizers-augmented SDT with anti-PD-L1 induces an anti-tumour response, which not only arrests primary tumour progression, but also prevents lung metastasis. Furthermore, the combined treatment strategy offers a long-term immunological memory function, which can protect against tumour rechallenge after elimination of the initial tumours. Therefore, this work represents a proof-of-concept combinatorial tumour therapeutics based on noninvasive tumours-therapeutic modality with immunotherapy.

---

[1] Department of Medical Ultrasound, Shanghai Tenth People's Hospital, Ultrasound Research and Education Institute, Tongji University School of Medicine, Shanghai 200072, P. R. China. [2] State Key Laboratory of High Performance Ceramics and Superfine Microstructure, Shanghai Institute of Ceramics, Chinese Academy of Sciences, Shanghai 200050, P. R. China. [3] Department of Gastroenterology, Shanghai Tenth People's Hospital, Tongji University School of Medicine, Shanghai 200072, P. R. China. Correspondence and requests for materials should be addressed to K.Z. (email: zhang1986kun@126.com) or to H.X. (email: xuhuixiong@126.com) or to Y.C. (email: chenyu@mail.sic.ac.cn)

A groundswell of research has recently been conducted on cancers and their complex interactions with the immune system, yielding a deeper understanding of how cancer progresses and thus offering new ways to stop it[1,2]. Based on the success of anti-CTLA4[3,4] and PD-L1/PD-1[5–9] treatment, the survival rates of patients with various types of haematologic and solid malignancies are improving[10]. Ira Mellman et al. have proposed that cancer immunotherapy has come of age[11]. However, these antitumour responses only occur in a subset of patients and cause specific immunotoxicity in some patients[12,13], significantly hindering their broad clinical use. How to safely and effectively induce and enhance the determinants of immune response is the crucial but challenging factor for substantial therapeutic benefits[13].

The combination of checkpoint blockade with some promising therapeutic modalities (i.e., chemotherapy[13], radiofrequency ablation (RFA)[14], radiotherapy[15–17], photothermal therapy (PTT)[18,19] and photodynamic therapy (PDT)[20]) has attained a synergistic effect on cancer treatment. However, all of these explored combinatorial therapeutic modalities suffer from their intrinsic critical issues, including high toxicity of chemotherapy, poor tissue-penetrating ability of phototherapies, invasive RFA and ionising radiotherapy, all of which inevitably cause low therapeutic efficiency and severe side effects, thus hindering their further clinical translations.

As one of the most representative noninvasive physical-irradiation sources, ultrasound (US) not only has a role in diagnostics in the clinic but also exerts a specific functionality in therapy. Sonodynamic therapy (SDT) as an emerging but representative US-based therapeutic modality for cancer noninvasive treatment features a high tissue-penetrating capability, non-ionising property, high controllability and low cost[21]. In the typical SDT process, US activates the sonosensitisers to generate reactive oxygen species (ROS) for inducing cancer-cell death via pathways of apoptosis and/or necrosis[21,22]. Preclinical animal models have documented that the debris from tumour cells induced by SDT process could serve as a source of tumour antigens to elicit host antitumour immunological effects[23]. In particular, such effects have also been validated in the preliminary clinical trial studies of breast-cancer patients[24,25]. Unfortunately, the extent of immune responses arising from SDT is not yet robust enough to efficiently prohibit tumour growth and metastasis.

Inspired by the intriguing therapeutic features of SDT and the intrinsic SDT-induced immune response, herein we establish a combined therapeutic strategy integrated with SDT, enhanced SDT-based immunotherapy and anti-PD-L1, which is successfully achieved based on the constructed immune-adjuvant and sonosensitisers co-loaded nano-liposomes as the nanosonosensitisers. The main components of SDT-based composite nanosonosensitisers are all U.S. Food and Drug Administration (FDA)-approved agents, e.g., liposome as the carrier for encapsulating therapeutic agents, hematoporphyrin monomethyl ether (HMME) as the US-responsive sonosensitiser and imiquimod (R837)-a toll-like receptor-7 (TLR7) agonist[26] as the immune adjuvant. The tumour-associated antigens in situ derived from SDT of the primary tumours have vaccine-like functions together with an immune adjuvant, which elicits an immune response via promoting dendritic cell (DCs) maturation and cytokine secretion, killing tumour cells. In particular, after combination with anti-PD-L1 checkpoint blockade, the systematic antitumour immune responses, including levels of tumour-infiltrating CD8[+] lymphocytes, are shown to be increased. Based on the underlying principle of antitumour immunotherapy, this combined therapeutic strategy has been demonstrated to not only suppress the primary tumours treated with SDT but also mitigate the

progression of tumour metastasis in 4T1 breast cancer and CT26 colorectal cancer murine models (Fig. 1), including subcutaneous, orthotopic and artificial whole-body metastasis tumour models. Furthermore, the combined tumour immunotherapy strategy also offers a long-term immunological memory function, which can protect against tumour rechallenge after elimination of the initial tumours. This work demonstrates the potential use of SDT in combination with checkpoint blockade in tumour treatment.

## Results

**Formulation and characterisation of HMME/R837@Lip.** Liposomes were used as nanoplatforms for co-encapsulating two hydrophobic molecules, i.e., sonosensitisers (HMME) and immune adjuvants TLR7 agonist (R837), via the typical reverse evaporation method (designated as HMME/R837@Lip)[27] (Fig. 2a), as liposomes are endowed with high biocompatibility and have been approved for clinical use by FDA[28]. The obtained HMME/R837@Lip composite nanosonosensitisers were well-dispersed in aqueous solution and appeared as quasi-spheres with homogeneous sizes by transmission electron microscopy (TEM) observation (Fig. 2b). High dispersity of HMME/R837@Lip guarantees the following in vitro and in vivo performance evaluations and biomedical use. The average hydrodynamic diameter of these nanosonosensitisers is around 157.3 nm, as determined by dynamic light scattering (DLS) (Fig. 2c). The nanoscale size benefits the substantial penetration into tumours via the EPR effect, thus enabling immune-adjuvant-enhanced immunotherapy during SDT process. HMME/R837@Lip also features a favourable structural stability in physiological conditions, as demonstrated by the negligible size variation and low polydispersity index (PDI), even though the incubation time of nanosonosensitisers in phosphate-buffered saline (PBS) containing 5 mg/mL bovine serum albumin exceeded more than 7 days at 4 °C (Supplementary Fig. 1).

An evident characteristic absorption peak at ~398 nm assigned to HMME in the UV-vis absorption spectrum of HMME/R837@Lip indicates the successful encapsulation of HMME into nanoparticles (Fig. 2d), which can also be validated by Zeta potential variation from Lips to HMME@Lip (Fig. 2e) and the high-performance liquid chromatography (HPLC) results (Supplementary Fig. 2a). The successful co-loading of R837 was further confirmed by HPLC (Supplementary Fig. 2b), as well as the increased surface Zeta potential from HMME@Lip to HMME/R837@Lip (Fig. 2e). The loading efficiency (Supplementary Fig. 2c) and capacity (Supplementary Fig. 2d) of HMME and R837 in HMME/R837@Lip were controllable, which varied with the different initially feeding concentrations.

Nanosonosensitiser-assisted SDT typically induces the production of ROS, such as singlet oxygen ($^1O_2$), which kills cancer cells[21] (Fig. 2f). To monitor the ROS species during the SDT process, electron spin resonance (ESR) and a 1,3-diphenyliso-benzofuran (DPBF) assay were employed to analyze ROS generation, qualitatively and quantitatively[21,29]. We found obvious and strong ESR characteristic peaks of $^1O_2$ species using HMME/R837@Lip as the nanosonosensitisers under US irradiation, while no evident peaks were observed in the TEMP + US group (Fig. 2g). The $^1O_2$ production was also demonstrated by evaluating DPBF degradation caused by $^1O_2$ production from SDT (Fig. 2h). The efficient $^1O_2$ production suggests a desirable SDT-based therapeutic outcome against tumours.

**HMME/R837@Lip-augmented SDT for immune-system activation.** After the successful demonstration of efficient $^1O_2$ production derived from HMME/R837@Lip-mediated SDT, we tested the system in vitro. Initially, the interaction between

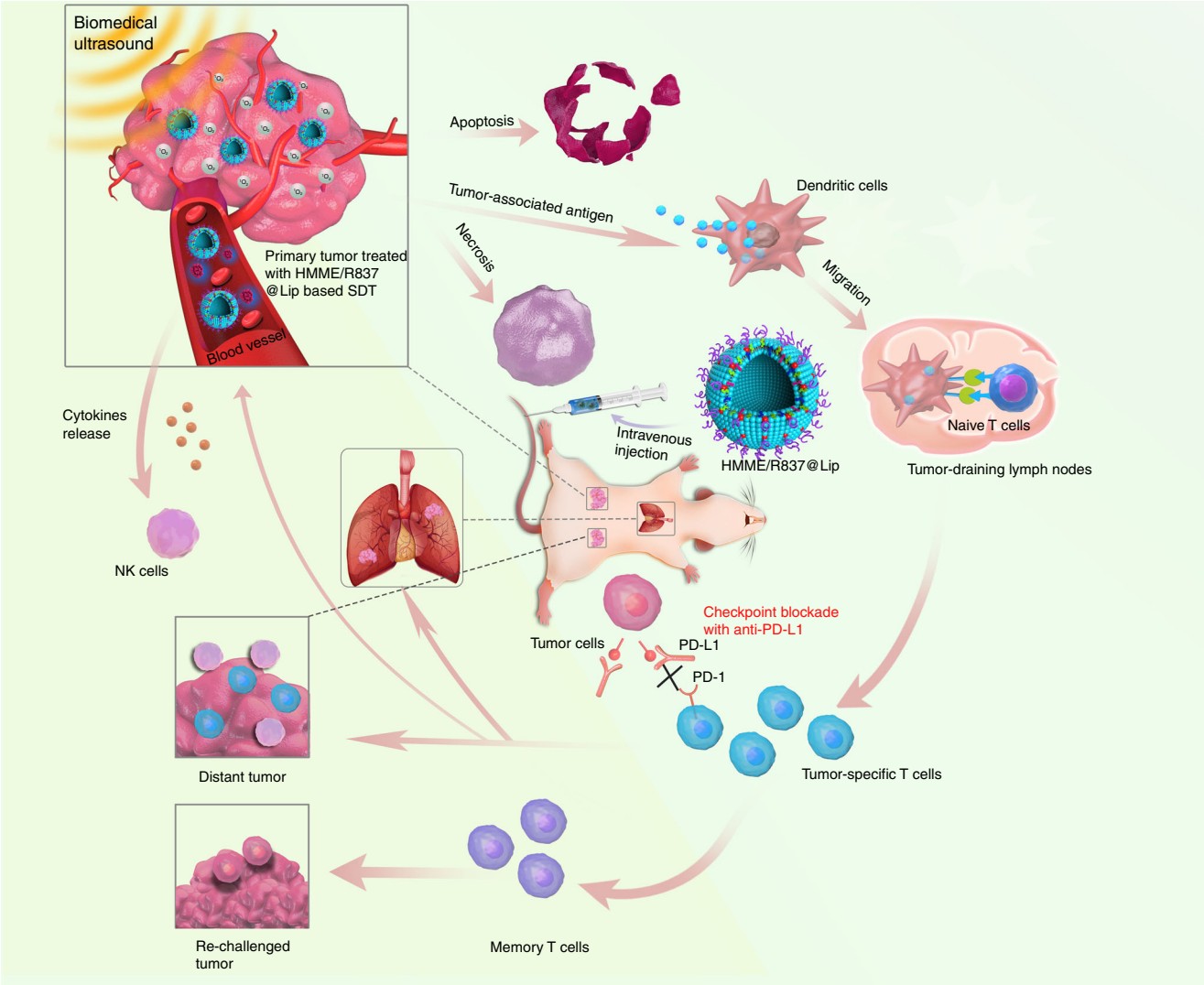

**Fig. 1** The design principle of nanosonosensitiser-augmented synergistic SDT and immunotherapy. Schematic illustration of antitumour immune responses induced by combined noninvasive SDT with immune-adjuvant-contained nanosonosensitisers and checkpoint blockade for effective cancer immunotherapy

HMME/R837@Lip and tumour cells (i.e., 4T1 breast-cancer cell) was explored. Most HMME/R837@Lip nanosonosensitisers were rapidly internalised by cancer cells within 4 h of co-incubation and retained in tumour cells over 24 h without an obvious efflux (<3%) (Supplementary Fig. 3a, b). This result was further confirmed by confocal laser scanning microscopy (CLSM) (Supplementary Fig. 3c). The efficient intracellular uptake and low efflux ensure high cellular accumulation of HMME/R837@Lip, enabling the intracellular SDT effect upon US irradiation.

To uncover the underlying mechanism of in vitro SDT using HMME/R837@Lip nanosonosensitisers to kill tumour cells, the intracellular ROS production level was detected by 2′-7′-dichlorofluorescein diacetate (DCFH-DA), which specifically targets and labels ROS[21,29]. As expected, both HMME@Lip and HMME/R837@Lip intracellularly produce large amounts of ROS after US irradiation, as shown by the strong intracellular green fluorescence in 4T1 cancer cells (Fig. 3a). Comparatively, R837 did not influence ROS production, as shown by the approximately identical fluorescence intensity in 4T1 cells treated with HMME@Lip and HMME/R837@Lip, respectively. Together with the aforementioned in vitro ESR results, we conclude that HMME/R837@Lip could intracellularly generate ROS under US

irradiation to induce toxic effects and further cause apoptosis and/or necrosis.

The sonotoxicity of HMME/R837@Lip against cancer cells was evaluated by CLSM (Fig. 3b). HMME/R837@Lip induced cancer-cell death (red fluorescence) under US irradiation, which was consistent with quantitative results of the cell-counting kit 8 (CCK-8) assay (Supplementary Fig. 4a, b).

To further demonstrate the function of the R837 component in HMME/R837@Lip as an immune adjuvant to induce immune response after SDT, flow-cytometry (FCM)-related analysis was conducted. As one of the most important antigen-presenting cells (APCs), DCs play crucial roles in innate and adaptive immunities[30]. Once exposed to antigens, the immature DCs engulf antigens and process them into peptides as they migrate to the nearby lymph nodes. Thereafter, the immature DCs undergo maturation and present the complex peptide to the native T cell. Hence, the immunological effects of HMME/R837@Lip towards bone-marrow-derived DCs were evaluated by analysing the upregulations of co-stimulatory molecules CD80/CD86 that are regarded as the representative markers for DC maturation[31].

A transwell system was employed to investigate this effect in vitro. As indicated in Fig. 3c, 4T1 cancer cell after various

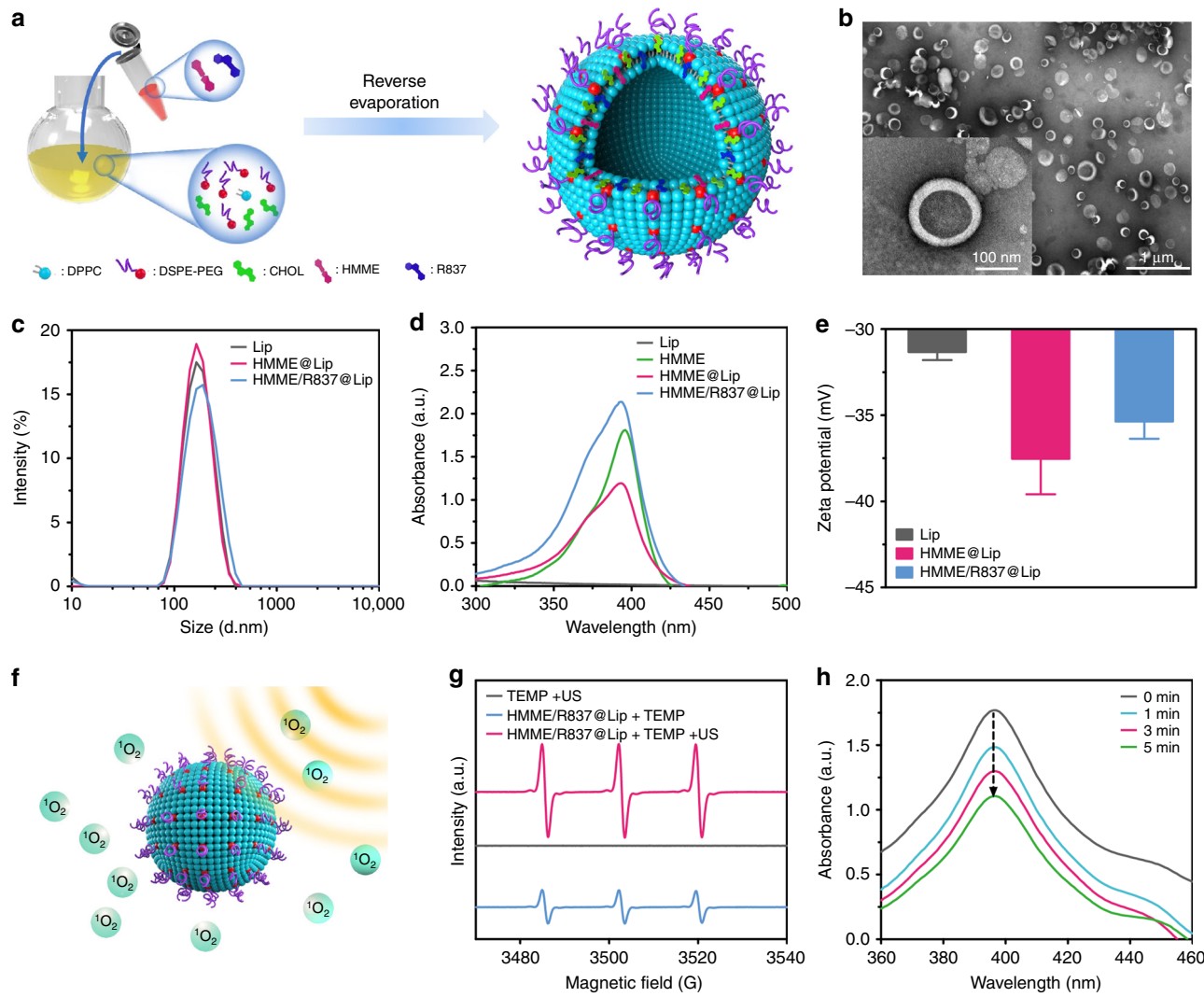

**Fig. 2** Synthesis and characterisations of the composite nanosonosensitisers. **a** Schematic illustration of the construction of HMME/R837@Lip nanosonosensitisers and their microstructures; **b** TEM image showing the quasi-spherical morphology of HMME/R837@Lip with high dispersity (scale bar = 100 nm); **c** hydrodynamic diameters of HMME/R837@Lip nanosonosensitisers in PBS as measured by DLS; **d** UV-vis absorbance spectra of Lip, HMME, HMME@Lip and HMME/R837@Lip, indicating the successful encapsulation of HMME into the nano-liposome; **e** zeta potential of Lip, HMME@Lip and HMME/R837@Lip, error bars are based on SD ($n = 3$); **f** scheme of US-triggered $^1O_2$ production as assisted by HMME/R837@Lip; **g** ESR spectra of HMME/R837@Lip with or without US treatment; **h** time-dependent DPBF absorption spectra in the presence of HMME/R837@Lip under US irradiation for varied durations

treatments and DCs were incubated in the upper and lower chamber, respectively. The extent of DC maturation and the related cytokine secretion were evaluated by FCM and enzyme-linked immunosorbent assay (ELISA), respectively. As HMME had certain dark toxicity against cancer cells, that is to say, even without US irradiation, some cancer cells co-cultured with HMME@Lip die (Fig. 3b; Supplementary Fig. 4a). Therefore, residual 4T1 cancer cells after incubation with HMME@Lip could slightly help to promote DC maturation. We found that tumour-cell debris after HMME/R837@Lip-augmented SDT could significantly promote DC maturation (CD80$^+$CD86$^+$ DCs) in comparison with HMME@Lip-assisted SDT (Fig. 3d, e). This result is attributed to the presence of R837 in HMME/R837@Lip because R837 alone also promotes DC maturation (22.5%).

Cytokine secretion is also a typical indication of immune responses[32,33]. Consistent with the DC maturation results, tumour-cell debris after HMME/R837@Lip-augmented SDT could act as an antigen to trigger the highest level of DC-secreted immune cytokines, such as IL-6 and TNF-α (Fig. 3f, g),

which further demonstrated that the immune adjuvant R837 could enhance the immune response. Therefore, tumour-associated antigens derived from tumour-cell residues after SDT, in combination with R837-containing nanoparticles as the immune-stimulating adjuvant, efficiently triggered DC maturation.

Encouraged by the in vitro results, HMME/R837@Lip nanosonosensitisers were further employed for in vivo anti-tumour evaluation (Fig. 4a). Initially, we explored the therapeutic ability of HMME/R837@Lip under US irradiation to induce localised cancer-cell apoptosis and/or necrosis in vivo by haematoxylin–eosin (H&E) staining and TdT-mediated dUTP nick-end labelling (TUNEL) assay. The histological results of tumours treated with HMME@Lip/HMME/R837@Lip but without US irradiation showed compact tumour cells with an intact structure. Comparatively, obviously separated and sparse tumour cells were found in tumours after HMME@Lip or HMME/R837@Lip-augmented SDT (Fig. 4b), indicating that a majority of tumour cells underwent apoptosis and/or necrosis. These results

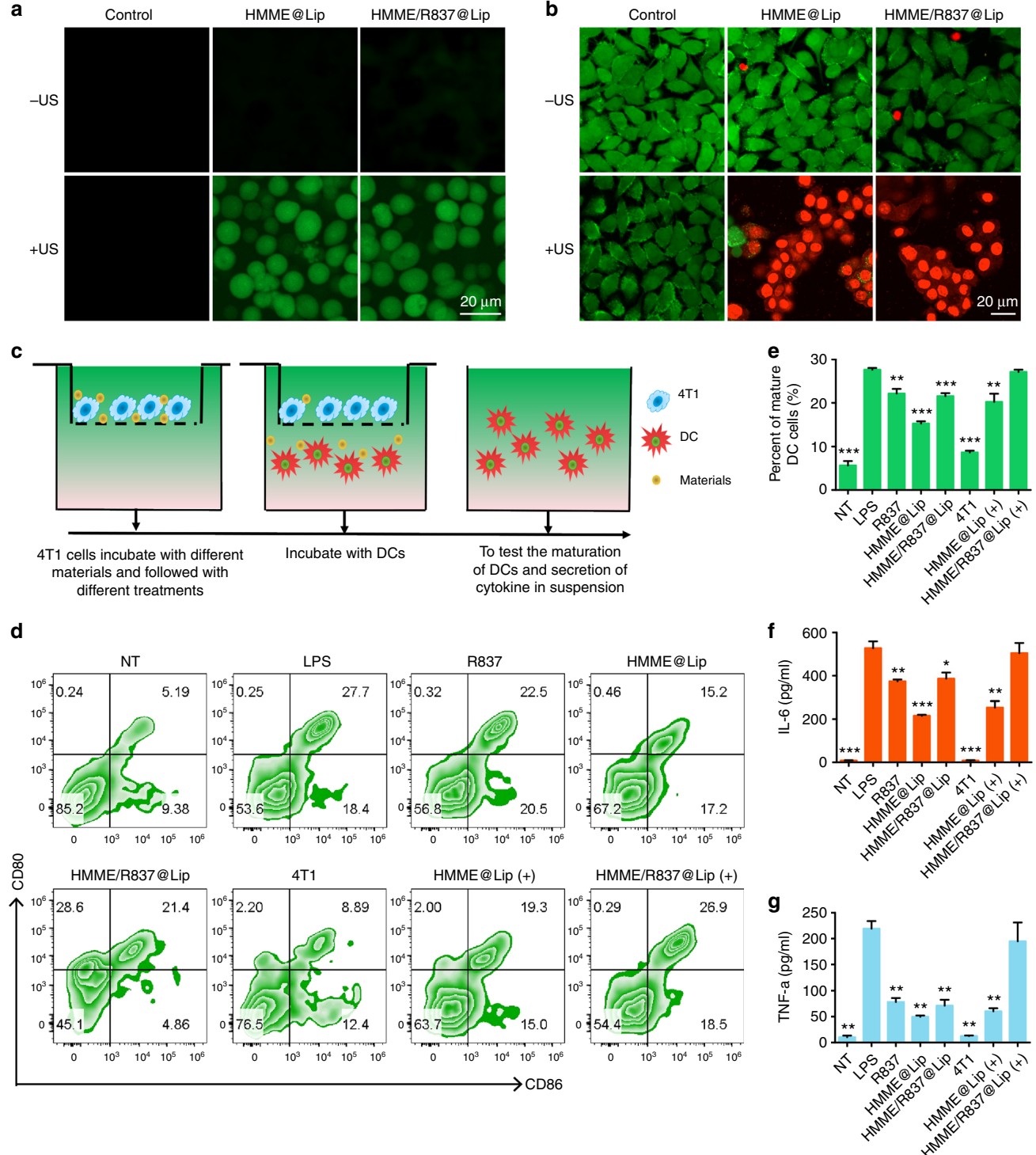

**Fig. 3** In vitro toxicity of HMME/R837@Lip upon US irradiation against 4T1 cancer cells and stimulation of in vitro immune response. **a** CLSM images of 4T1 cells stained with DCFH-DA after different treatments: control (without any treatment), HMME@Lip only, HMME/R837@Lip only, US only, HMME@Lip combined with US irradiation and HMME/R837@Lip combined with US irradiation (scale bar = 20 μm); **b** CLSM images of 4T1 cells stained with Calcein-AM and PI after various treatments: control (without any treatment), HMME@Lip only, HMME/R837@Lip only, US only, HMME@Lip combined with US irradiation and HMME/R837@Lip combined with US irradiation (scale bar = 20 μm); **c** the design scheme of the transwell system experiment. 4T1 cells and their residues were placed in the upper compartment and DCs were cultured in the lower compartment; **d** the percentage of mature DCs (CD11c⁺CD80⁺CD86⁺) was analysed by flow cytometry after different treatments for 20 h in the transwell system; **e-g** quantification of the level of DC maturation (**e**) and the secretion of IL-6 (**f**) and TNF-α (**g**) in DC suspensions. Data are expressed as means ± SD ($n = 3$). Statistical significances were calculated via Student's $t$ test, *$p < 0.05$, **$p < 0.01$ and ***$p < 0.001$

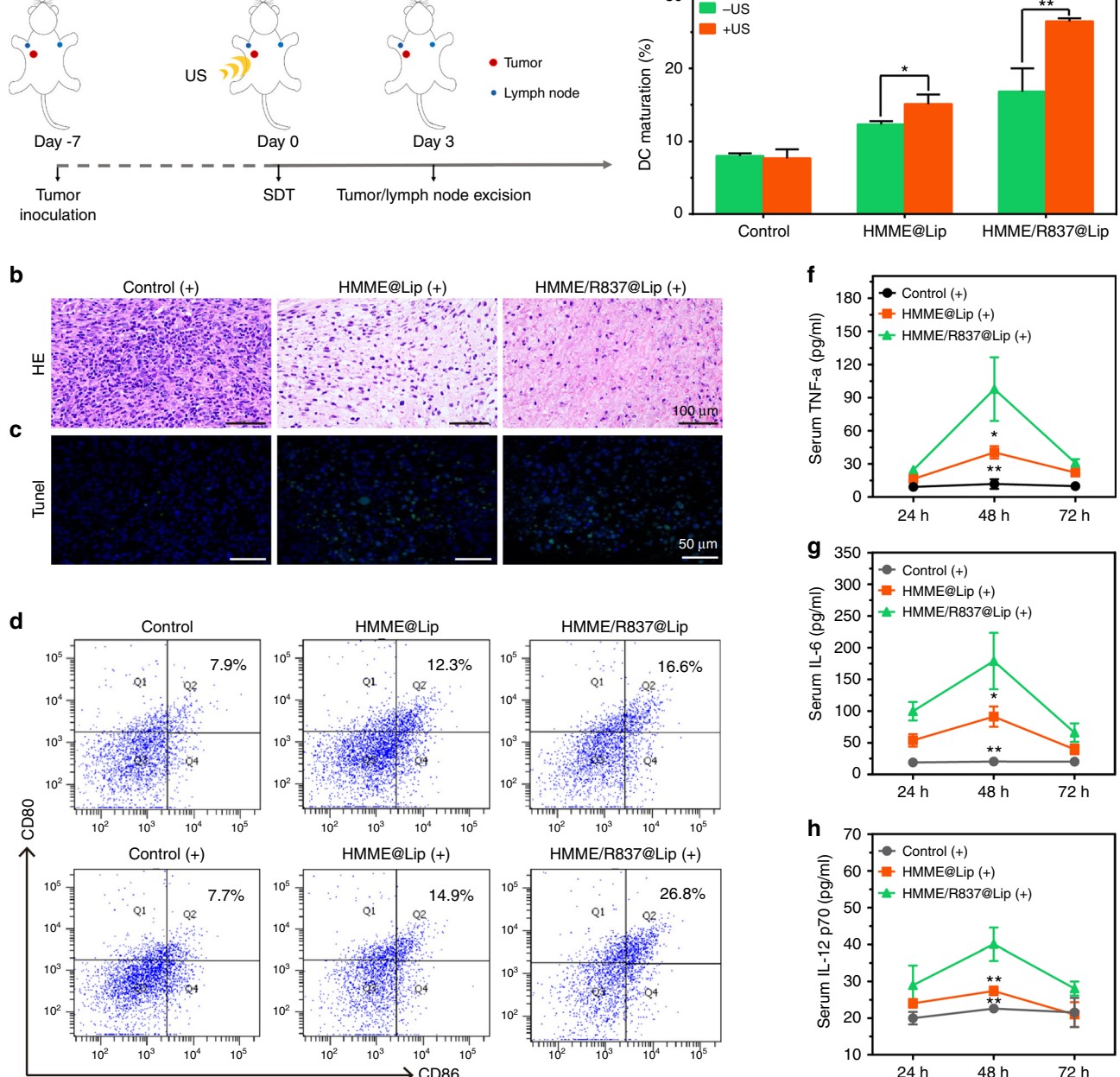

**Fig. 4** In vivo HMME/R837@Lip-augmented SDT for inducing cell apoptosis and/or necrosis, promoting DC maturation and stimulating the expression of proinflammatory cytokines. **a** Schematic illustration of the experiment design to assess the in vivo sonotoxicity and the immune responses as triggered by HMME/R837@Lip-augmented SDT; **b**, **c** in vivo apoptosis and/or necrosis of the tumour induced by HMME/R837@Lip-augmented SDT, as shown by H&E staining (scale bar = 100 μm) and TUNEL assay (scale bar = 50 μm); **d**, **e** DC maturation in the tumour-draining lymph nodes induced by HMME/ R837@Lip-augmented SDT on mice bearing 4T1 tumours, as assessed by flow cytometry after staining with CD11c, CD80, CD86 and live dead; **f**–**h** cytokine levels of TNF-α, IL-6 and IL-12p70 in sera from mice isolated from 24 to 72 h after HMME/R837@Lip-agumented SDT treatment. Data are expressed as means ± SD ($n = 3$). Statistical significances were calculated via Student's $t$ test, *$p < 0.05$, **$p < 0.01$

were further confirmed by the high percentage of TUNEL-positive cells (green fluorescence) in TUNEL assay (Fig. 4c).

Similar to the in vitro results, we further demonstrated that the integration of the R837 immune adjuvant endows HMME/ R837@Lip with a much stronger ability to promote DC maturation (Fig. 4d, e) and secretes immune cytokines in vivo (Fig. 4f–h), which enables the immunotherapy against solid tumours combined with SDT. In detail, SDT with HMME/ R837@Lip induced high levels of DC maturation (26.8%), which was much higher than the composite nanosonosensitisers alone (16.6%) and HMME@Lip (14.9%). Therefore, after the tumours

were destroyed by SDT, particularly in the presence of an immune adjuvant, tumour-associated antigens shedding off the tumour debris could be transported into the nearby lymph nodes and processed by DCs to effectively simulate DC maturation.

Calreticulin (CRT) exposed on the cell surface is a distinct biomarker of immunogenic cell death (ICD)[34]. Once on the tumour-cell surface, CRT acts as an "eat me" signal, stimulating macrophages and DCs to engulf the dying cells and their apoptotic debris[35]. Thus, we assessed levels of CRT exposure on the 4T1 tumours treated with HMME/R837@Lip plus irradiation. As shown in Supplementary Fig. 5, HMME/R837@Lip-augmented

SDT treatment significantly induced CRT expression, and that was in line with the results of previous studies[23,36]. In addition, the variations of cytokines (e.g., TNF-a, IL-6, and IL-12p70) in the serum of the 4T1-tumour-bearing mice after treatment indicated that although both HMME/R837@Lip (−) and HMME@Lip (+) increased secretion of the proinflammatory cytokines, the secretions induced by HMME/R837@Lip (+) were much higher, which were favourable for triggering the antitumour immunological responses. However, 72 h after SDT treatment, all three proinflammatory cytokine levels rapidly dropped to nearly baseline levels, suggesting that inflammation caused by HMME/R837@Lip was only an acute response. Although the administration of R837 via the intravenous injection route might induce severe side effects (e.g., cytokine storms), we did not observe the notable cytokine-storm-like side effect. All mice behaved normally after treatments with i.v. injected HMME/R837@Lip without accidental death, as a result of the used nanoparticle-delivery system (liposome).

**SDT plus PD-L1 blockade for suppression of distant tumours.** The major cause of cancer-induced deaths, the occurrence of tumour metastasis[9,37], is often accompanied by resistance to conventional therapies, including surgical operation, radiation and chemotherapy[38,39]. Therefore, it has been generally recognised that the desirable cancer treatment should not only destroy the primary tumour but also recognise, suppress and remove any residual cancer cells at the site of the metastasis. PD-1/PD-L1 checkpoint blockade has been proven to promote antitumour immunities by inhibiting cytotoxic T-lymphocyte exhaustion with the exciting clinical results. Especially, after combination with other treatment protocols, the antitumour outcome is significantly enhanced. Thus, PD-L1 blockade was herein employed to further enhance the antitumour immunotherapeutic efficacies as generated by HMME/R837@Lip-augmented SDT.

To evaluate the synergistic tumour-inhibition efficiency of the integrated PD-L1 blockade and HMME/R837@Lip-augmented SDT, a bilaterally bearing 4T1 tumour model was employed. The experimental procedure is shown in Fig. 5a, wherein the second tumour inoculated on the left chest after 6 days post the first tumour inoculation on the right chest was set as an artificial model of metastasis. Before therapeutic experiment, in vivo accumulation of Cy5.5-labelled HMME/R837@Lip in tumour tissue was initially explored with an in vivo fluorescence imaging system (IVFIS). It revealed that these HMME/R837@Lip nanosonosensitisers were accumulated into 4T1 tumours after i.v. administration (Fig. 5b, c), which was contributed by the long blood-circulation duration and high stability in favour of the EPR effect. Even after up to 48 h, HMME/R837@Lip nanoparticles were retained in the tumour (Fig. 5c–e), enabling the repeated US treatment after a single injection.

The tumour-bearing mice were treated with US irradiation twice at 12 h and 24 h after each i.v. injection of the nanosonosensitisers, respectively. After the second irradiation, the mice were intraperitoneally injected with anti-PD-L1 antibody at a dose of 75 μg/mouse, and thus the equivalent injections were also performed on the 6th and 8th days. Treatment results of different groups against the primary and mimic distant tumours are summarised in Fig. 5f–i and Supplementary Fig. 6a, b. Anti-PD-L1 alone exhibited a little effect on the inhibition of primary and distant tumours. Although HMME/R837@Lip-augmented SDT alone could suppress the primary tumour growth, it failed to exert the influences on the distant tumour (mimic metastasis). Notably, HMME/R837@Lip-augmented SDT combined with anti-PD-L1 blockade not only almost completely eradicated the primary 4T1 tumour (VRR, 95%) but also significantly

suppressed the distant tumour growth (VRR, 83%) (Fig. 5f–h). The variation trend of tumour volume was consistent with that of tumour weight, as indicated in Fig. 5g–i, wherein HMME/R837@Lip-augmented SDT combined with anti-PD-L1 blockade acquired the largest reduction of tumour weight, in both primary and distant tumours. In addition, no abnormal weight or temperature changes of mice were observed in the HMME/R837@Lip-SDT plus anti-PD-L1 group (Fig. 5j, k), indicating the high therapeutic biosafety of the combined cancer SDT and immunotherapy.

The aforementioned anti-metastasis evaluation was implemented on subcutaneously xenografted tumour. Furthermore, orthotopic 4T1 tumour models mimicking human breast cancer were employed to assess the anti-metastasis efficiency of HMME/R837@Lip-augmented SDT plus PD-L1 blockade. The schematic depicting the experimental procedures is shown in Fig. 6a. In detail, a bilateral orthotopic 4T1 tumour model was initially established by injecting cancer cells into mammary fat pads of female BALB/c mice. The first tumour was inoculated into the right abdominal mammary fat pad as the primary tumour. Six days later, the second tumour was inoculated into the left abdominal mammary fat pad as an artificially distant metastatic tumour. Subsequently, all the following procedures, including grouping and the corresponding treatment protocols, were the same as those designed in the aforementioned subcutaneous tumour model experiment.

The biodistribution of HMME/R837@Lip nanosonosensitisers was also evaluated, and a high accumulation of nanoparticles in the orthotopic tumour model was recorded (Fig. 6b, c). With regard to antitumour, the results on the orthotopic tumour model were approximately the same as those of the subcutaneously xenografted tumour model. The combination of HMME/R837@Lip + US and anti-PD-L1 blockade resulted in the robust antitumour ability, due to the three synergistic contributions from HMME/R837@Lip-augmented SDT, R837-enhanced immune response, and anti-PD-L1 blockade-mediated immunotherapy (Supplementary Fig. 7). Consequently, this distinctive treatment strategy resulted in the largest suppression rate and weight reduction of both primary orthotopic (Fig. 6d–f) and mimic distant tumours (Fig. 6g–i).

In addition to the bilateral orthotopic tumour model alone, the efficacy of the combined SDT/immunotherapy strategy was also verified by an aggressive whole-body spreading tumour model. This model was established by i.v. injection of 4T1 cancer cells expressing firefly luciferase (fLuc-4T1) into unilateral orthotopic tumour-bearing mice before the primary tumours were eliminated by HMME/R837@Lip-augmented SDT and enhanced immunotherapy deriving from R837 and anti-PD-L1 blockade. IVFIS was introduced to track the migration pathway of i.v. injected fLuc-4T1 tumour cells (Fig. 6j). The bioluminescence imaging revealed that cancer metastasis started on the 15th day after i.v. injection of the fLuc-4T1 cells into the mice treated with PBS, anti-PD-L1 and HMME/R837@Lip + US. When prolonging of the feeding duration, augmented bioluminescence signals were detected in almost all mice in these three groups on the 19th day. In contrast, the mice treated with HMME/R837@Lip-augmented SDT plus anti-PD-L1 exhibited negligible bioluminescence signals, indicating the substantially suppressed tumour metastasis. To directly evaluate the lung metastasis, lungs in all groups were collected. The gross appearance of lung nodules revealed that compared with three control groups, HMME/R837@Lip + US + anti-PD-L1 group exerted a significant effect on preventing lung metastasis (Fig. 6k). The quantitative results also confirmed that the combined treatment strategy substantially reduced lung nodules in comparison with the PBS group, i.e., $5.6 \pm 4.5$ lung nodules vs. $30.8 \pm 14.1$ lung nodules (Fig. 6l). The lungs were

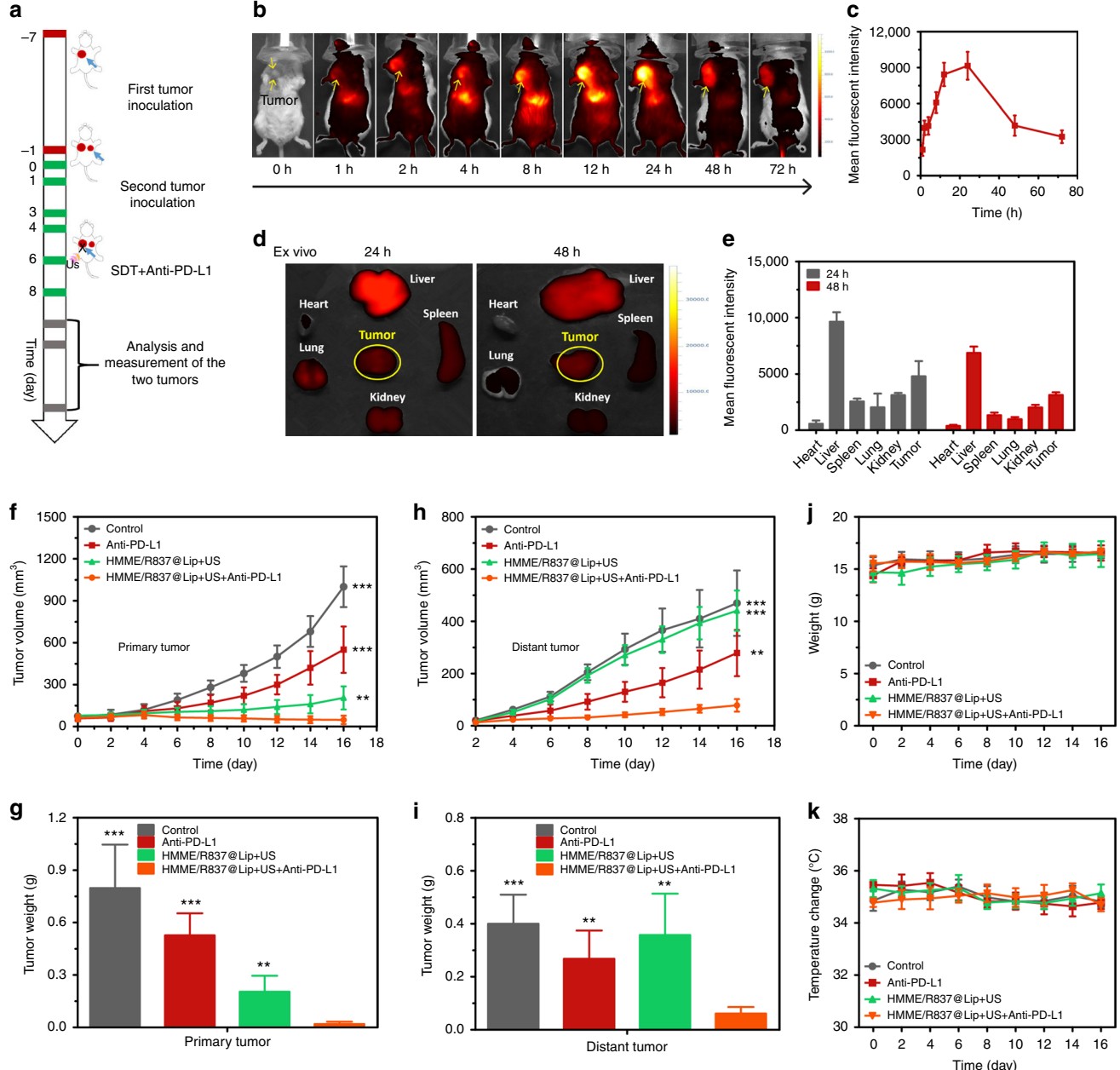

**Fig. 5** Synergistic nanosonosensitiser-augmented SDT and immunotherapy for in vivo suppression of mimic distant tumours in a subcutaneous tumour model. **a** Schematic showing the experiment design for in vivo evaluations. Mice bearing 4T1 subcutaneous tumours on both sides were used in this experiment. Tumours on the right side of the check were designed as "primary tumour" for SDT, and those on the left side were set as "mimic distant tumours" without SDT; **b** in vivo fluorescence images to reveal the biodistribution of HMME/R837@Lip post i.v. injection into 4T1-tumour-bearing mice at the indicated time points. Yellow arrows indicate tumours; **c** the accumulation curve of HMME/R837@Lip in the tumour tissue by measuring the fluorescence intensity of tumours at different time points post i.v. injection, error bars are based on SD ($n = 3$); **d** the ex vivo fluorescence image of major organs and tumour dissected from the mouse 24 and 48 h post injection and (**e**) quantification analysis of the tissue content of HMME/R837@Lip by testing the corresponding fluorescence intensity; data are expressed as means ± SD ($n = 3$); **f**, **h** primary (**f**) and distant (**h**) tumours growth curves of different groups of tumour-bearing mice after various treatments as indicated in the figure. Error bars are based on SD ($n = 5$); average weights of **g** primary and **i** distant tumours at the end of treatment; time-dependent body temperature (**j**) and weight (**k**) surveillance of mice ($n = 5$) after different treatments. Statistical significances were calculated via Student's $t$ test, $^{**}p < 0.01$ and $^{***}p < 0.001$

further sectioned and stained with H&E for pathological analysis. As shown in Fig. 6m, the lungs were occupied by tumours in the PBS, anti-PD-L1 and HMME/R837@Lip + US groups, while only very few lung nodules were found in the combined treatment group, indicating that the combined SDT/immunotherapy strategy accommodated a robust ability of anti-lung metastasis.

In order to reveal the biosafety of this combined therapeutic strategy, the potential harmful effect to normal organs induced by

the intensified immune therapeutic effect was evaluated. The serum biochemistry assay was conducted on the tumour-bearing mice experiencing different treatments. It was recorded that all the measured indexes of the treated groups were not significantly different from those of the healthy control (Supplementary Fig. 8), indicating that such a boosted antitumour immunity of anti-PD-L1 plus HMME/R837@Lip-augmented SDT was tolerable by mice. No evident damage to the major organs (i.e., the

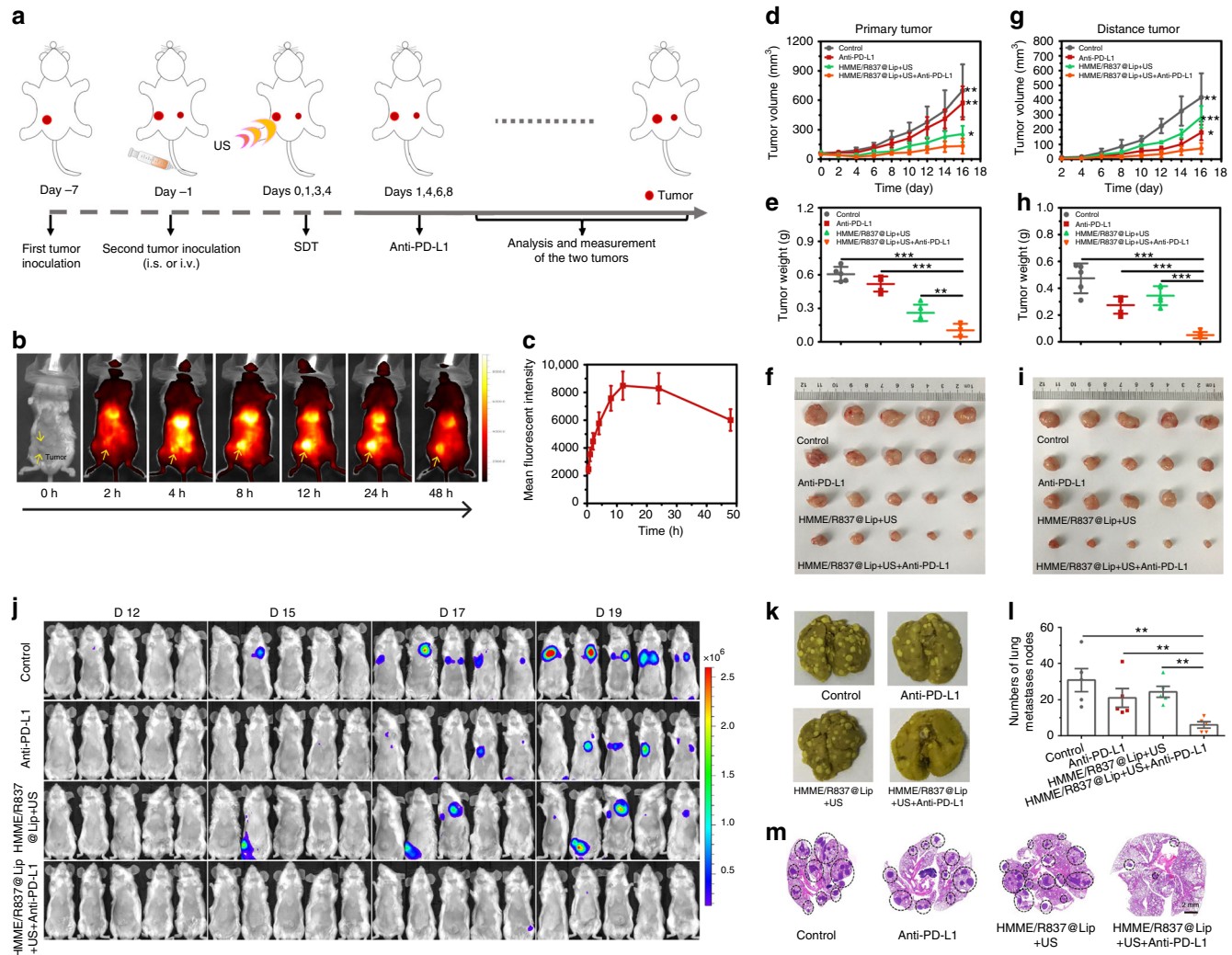

**Fig. 6** Antitumour effect of HMME/R837@Lip-augmented SDT plus anti-PD-L1 immunotherapy in orthotopic tumour models. **a** Schematic illustration of HMME/R837@Lip-augmented SDT and anti-PD-L1 combination therapy to inhibit tumour growth at the distant sites on orthotopic tumour models; **b** in vivo fluorescence images to study the biodistribution of HMME/R837@Lip post i.v. injection in 4T1 orthotopic tumour-bearing mice at the indicated time points. Yellow arrows indicate tumours; **c** the accumulation curve of HMME/R837@Lip in the tumour tissue by measuring the fluorescence of tumours at different time points post i.v. injection, error bars are based on SD ($n = 3$); **d–i** primary (**d**) and distant (**g**) tumour-growth curves of different groups of orthotopic tumour-bearing mice after various treatments as indicated in the figure. Error bars are based on SD ($n = 5$); average weights of primary (**e**) and distant (**h**) tumours at the end of treatments; photographs of excised primary (**f**) and distant (**i**) tumours at the end of treatments; **j** in vivo bioluminescence images tracking the spreading and growth of i.v. injected fLuc-4T1 tumour cells in the mice after different treatments; **k** representative photographs to show the gross appearance of tumour nodules in the lungs; **l** the numbers of lung nodules were counted under anatomy microscope. Values are means ± standard error (SE) ($n = 5$); **m** representative haematoxylin and eosin staining analysis of the lung metastasis. Scale bar = 2 mm. Statistical significances were calculated via Student's $t$ test, *$p < 0.05$, ** $p < 0.01$ and ***$p < 0.001$

heart, liver, lung, spleen and kidney) also confirmed the histocompatibility of anti-PD-L1 plus HMME/R837@Lip-augmented SDT (Supplementary Fig. 9). Similar to the experiments using a subcutaneous xenografted tumour model, no abnormal temperature or body-weight changes (Supplementary Fig. 10a, b) were observed in mice treated with the combined treatment. Together, our results demonstrate that the combined therapeutic strategy integrating HMME/R837@Lip-augmented SDT with anti-PD-L1 holds great potential in synergistic cancer immunotherapy.

**Mechanism of systematic antitumour immune responses.** To understand the underlying mechanism of the antitumour effects triggered by HMME/R837@Lip-augmented SDT plus PD-L1

blockade, immune cells in the mimic distant tumours were assessed on the 11th day after the first treatment using a bilaterally orthotopic model (Fig. 7a–d). The percentage of CD45[+] leucocytes in tumours was evaluated, because CD45[+] molecules are the well-recognised and common leucocyte-specific antigen[40]. The results showed that the proportion of CD45[+] leucocytes in the HMME/R837@Lip + US plus anti-PD-L1 group was 38 ± 5.9%, with a significantly increased magnitude of 54% compared with PBS control group (24.6 ± 2.0%) (Fig. 7a–d). Specifically, the absolute percentage of CD8[+] cells in the combined treatment group occupies 18.6% with an improvement by nearly 2.7 times in comparison with the PBS group (6.7%) (Fig. 7b–d). There was also a slightly and statistically insignificant increase in the percentage of NK cells in the distant tumours treated with HMME/R837@Lip + US + anti-PD-L1, while no evident effect on B cells

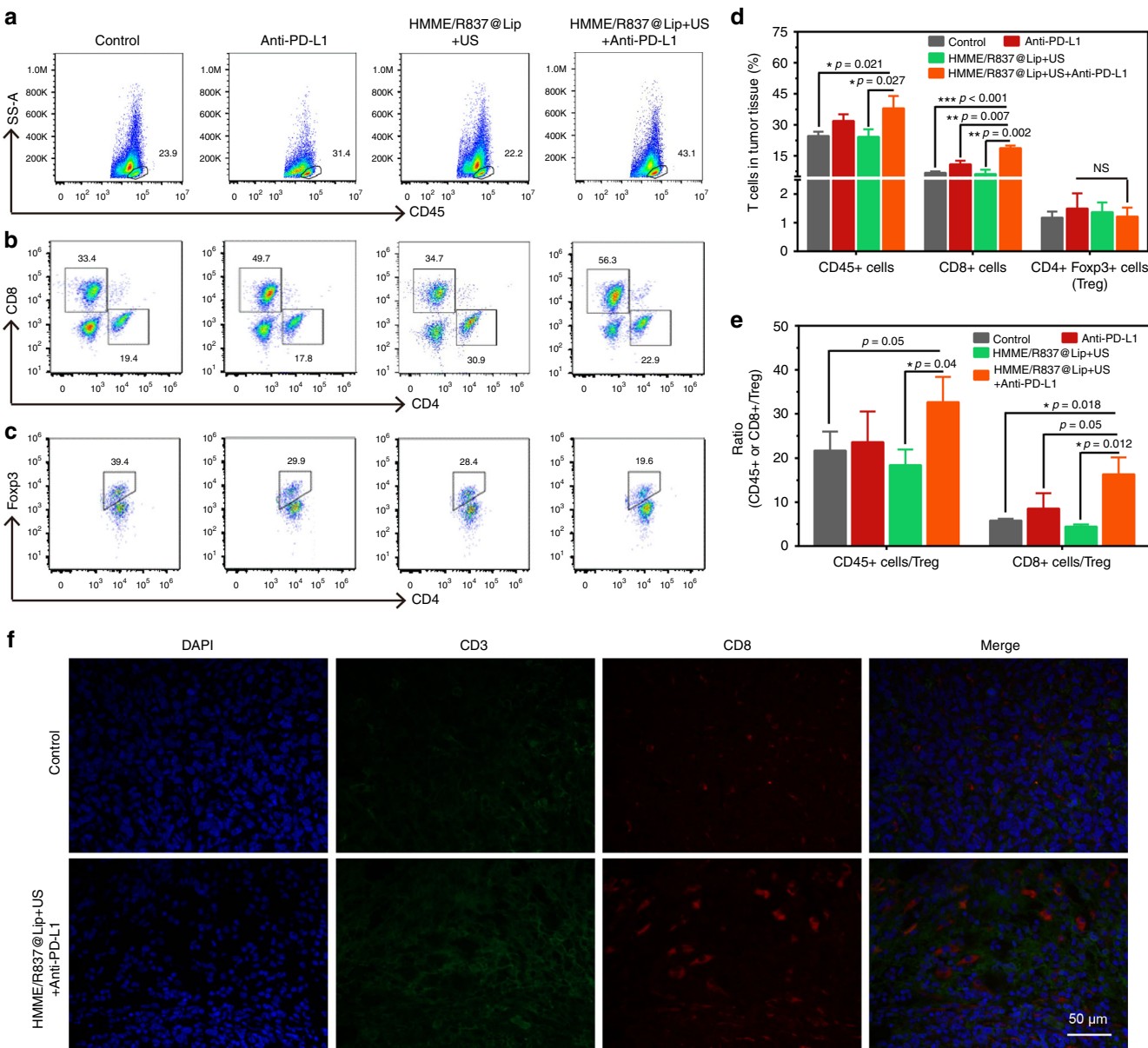

**Fig. 7** HMME/R837@Lip-augmented SDT plus anti-PD-L1 therapy activating systematic antitumour immunity. **a–c** Representative flow-cytometry plots showing the tumour-infiltrating leucocyte cells, including CD45+ cells (CD45+), CD8+ T cells (CD45+CD3e+CD8+) and CD4+ FoxP3+ T cells (CD45+CD3e+CD4+ FoxP3+) in mimic distant tumours; **d** absolute quantification of the CD45+ cells, CD8+ T cells and CD4+ FoxP3+ regulatory T cells (Treg) in mimic distant tumours (gating on CD3+ cells), error bars are based on SD ($n = 3$); **e** CD45+ cells: Treg ratios and CD8+cells: Treg ratios in the distant tumours after various treatments to remove the first tumour. Values are mean ± SD ($n = 3$); **f** representative CLSM images of the mimic distant tumours after immunofluorescence staining (scale bar = 50 μm). Statistical significances were calculated via Student's $t$ test or Mann–Whitney U test. NS not significant. *$p < 0.05$, **$p < 0.01$ and ***$p < 0.001$

in the combined treatment protocol was observed (Supplementary Fig. 11). These results indicate that this systematic antitumour immunity is potentially contributed by the cell-mediated immunity.

Based on the level of Foxp3 marker, CD4+ T cells can be classified into two types, i.e., effective T cells (Teffs, CD4+Foxp3+) and regulatory T cells (Tregs, CD4+Foxp3−). Teffs can promote immune responses, while Tregs hamper the antitumour immune responses[41,42]. The leucocytes in the mimic distant tumours were collected and co-stained with CD3, CD4 and Foxp3 for further analysis. Although CD4+ cells were substantially enriched in all four groups (Supplementary Fig. 11), the percentages of

immunosuppressive Tregs in tumours post combined treatment did not significantly decrease (Fig. 7c, d). However, the ratios of CD45+, CD8+ to Tregs in this combined therapeutic group were much higher than those in other three groups (Fig. 7e), which might also help to increase CD8+ T cells activity.

The antitumour immunity elicited by HMME/R837@Lip-augmented SDT plus anti-PD-L1 was further verified by the immunofluorescence assay. It has been found that HMME/R837@Lip with US irradiation plus PD-L1 blockade instigated CD3+T cells infiltration into the distant tumours, whereas few CD3+T cells were observed in the tumours of the PBS group (Fig. 7f). In addition, most tumour-infiltrating CD3+T cells were

CD8$^+$, which validated again that HMME/R837@Lip-augmented SDT plus anti-PD-L1 significantly promoted CD8$^+$T cells infiltration into tumours.

**Anticancer activity in the murine colorectal cancer models.** Based on the aforementioned results, the combined HMME/R837@Lip-augmented SDT with anti-PD-L1 therapy could elicit a strong and effective antitumor immune response in the 4T1 breast-cancer models. Therefore, we wondered whether such a combined tumour immunotherapy strategy could potentially be used for other tumour types. To further explore the results, murine colorectal CT26-cancer cell was employed in this work. We initially verified the intracellular ROS production (Supplementary Fig. 12) and sonotoxicity of HMME/R837@Lip against CT26-cancer cells (Supplementary Fig. 13) by in vitro experiments. For in vivo experiments, the bilateral subcutaneous CT26 model was used and the treatment plan was the same as the aforementioned protocol, as shown in Fig. 8a. The tumour sizes on both sides (Fig. 8b, c) and body weight (Fig. 8d) were closely monitored. It showed that CT26 tumour-bearing mice failed to efficiently respond to a single anti-PD-L1 blockade (Fig. 8b, c). Although HMME/R837@Lip-based SDT could partially delay the growth of the primary tumours, it exerted limited effects on the mimic distant tumours (Fig. 8c). However, when we combined HMME/R837@Lip-augmented SDT with anti-PD-L1, both the primary and distant tumours were significantly inhibited (Supplementary Fig. 14). We also found that four out of six mice survived for 50 days after this combined tumour immunotherapy, compared with mice in the other three control groups that all died within 16–28 days (Fig. 8e). The survived four mice in the combined treatment group were killed at day 50 for careful necropsy, and no visually noticeable metastatic tumours were discovered. All the mice behaved normally, and no abnormal body-temperature (Supplementary Fig. 15) or body-weight changes (Fig. 8d) were observed in HMME/R837@Lip-SDT plus anti-PD-L1 therapy group, once again indicating the absence of seriously systemic toxicity. All these results indicate that our findings with the conducted immunotherapy strategy in the 4T1 cancer models can be extended to other types of tumours.

Immunological memory response is a well-known feature of the adaptive immunities, and it has a crucial role in protecting organisms against the second pathogen attack[43,44]. That is to say, when we are exposed to the second encounter of a remember pathogen, the memory T cells could respond rapidly, inciting a faster and stronger immune response than the first time our immune system responds. Therefore, we further assessed the immune-memory effects of our combined tumour immunotherapy strategy (Fig. 8a) by rechallenging mice with the secondary inoculation of CT26-cancer cells on day 30, which was the 22nd day[19,45] after SDT plus anti-PD-L1 treatment to remove the first tumour. At the same time, eight age- and sex-matched native mice were inoculated with the same number of CT26 cells to help to identify the results. It showed that, the growth of re-inoculated tumours in the combined treatment group was inhibited in comparison with that of the native group (Fig. 8f; Supplementary Fig. 16). In total, 75% (6/8) of the mice after combined treatment were resistant to rechallenge. Comparatively, all of the control mice developed tumours and died within 26 days after inoculation (Fig. 8g). These results demonstrated that long-term immune-memory effects were generated by HMME/R837@Lip-augmented SDT plus anti-PD-L1 blockade treatment.

Finally, we attempted to further identify the possible mechanism of the robust immune memory post combined tumour immunotherapy by analysing the related cytokines (Fig. 8h) and T cells (Fig. 8i–k). On day 29 before the mice were rechallenged with the secondary tumours, spleens of mice after combined treatments were harvested and evaluated to determine the changes of naive, central memory (TCM) and effector memory T cells (TEM). Intriguingly, FCM data showed that HMME/R837@Lip-augmented SDT plus anti-PD-L1 antibody treatment for CT26 tumour-bearing mice led to an obvious shift of naive and TCM CD8$^+$ T cells towards TEM phenotype, compared with the control group. It has already been suggested that TCM mainly residing in the secondary lymphoid tissues only provides immunities after courses of expansion, differentiation and migration; while TEM, which locates in both lymphoid and non-lymphoid tissues, could induce immediate protections by producing multiple cytokines (i.e., TNF-α and IFN-γ) upon the second antigen encounter[46,47]. Therefore, the ability of the combined therapy to shift naive and TCM CD8$^+$ T cells towards a TEM phenotype further suggests the potential of generating more-potent antitumour immune effects. However, for the CD4$^+$ population, although the percentages of naive T cells and TCM of the combined treatment group were lower than those of the control group, there were no statistic differences in TEM levels between two groups. That was in line with the previous results that CD8$^+$ cells were the predominant T-cell subset that was responsible for the strong anticancer immune responses[45]. Furthermore, 7 days after the second tumour was introduced, serum cytokines, including IFN-γ and TNF-α, that play vital roles in cellular immunity against cancer, were analysed by ELISA. Our results show that they were significantly improved for mice treated with combined tumour immunotherapy, indicating the establishment of strong antitumour immune responses (Fig. 8h).

## Discussion

In order to overcome critical issues of traditional cancer-therapeutic modalities, great efforts have been devoted to some noninvasive therapeutics, such as SDT[21,29,48]. SDT was highlighted to effectively induce cancer-cell death and suppress tumour growth in diverse preclinical tumour models (e.g., breast cancer[21], brain glioma[29] and pancreatic cancer[22]), and even clinical reports[24,25]. HMME/R837@Lip nanosonosensitisers killed cancer cells by inducing apoptosis and/or necrosis and further stimulated the immune system to activate the adaptive immune responses via HMME-augmented SDT and immune-adjuvant R837-enhanced immune response. Also, the combination of HMME/R837@Lip-augmented SDT and PD-L1 checkpoint blockade effectively eradicated/suppressed growth of the primary tumour, and simultaneously evoked the systemic anticancer immunity. Furthermore, this combined treatment strategy can provide strong long-term immune-memory effects to prevent tumour reoccurrence.

With a glimpse at the recent advances of tumour immunotherapy, the combination paradigms between conventional cancer-treatment modalities (e.g., RFA[14], PDT[49] and PTT[50]) with CTLA-4 or PD-1/PDL-1 could maximise benefits of cancer immunotherapies. Compared with pre-existing combined immunotherapeutic strategies, this established strategy of nanosonosensitiser-augmented SDT integrated with anti-PD-L1 offers distinct advantages. First, in comparison with RFA that is minimally invasive in the clinic, SDT is a noninvasive treatment protocol[51]. Based on rational design, repeated US irradiations are applicable after one injection of nanosonosensitisers, which can improve the compliance of the patient once the combined treatment achieved clinical translation. Second, US irradiation in SDT is more preferable and applicable than light irradiation in PDT/PTT, because US as the irradiation source can acquire a much higher tissue-penetrating depth than light, enabling the treatment of deep-seated tumours. Finally, compared with the

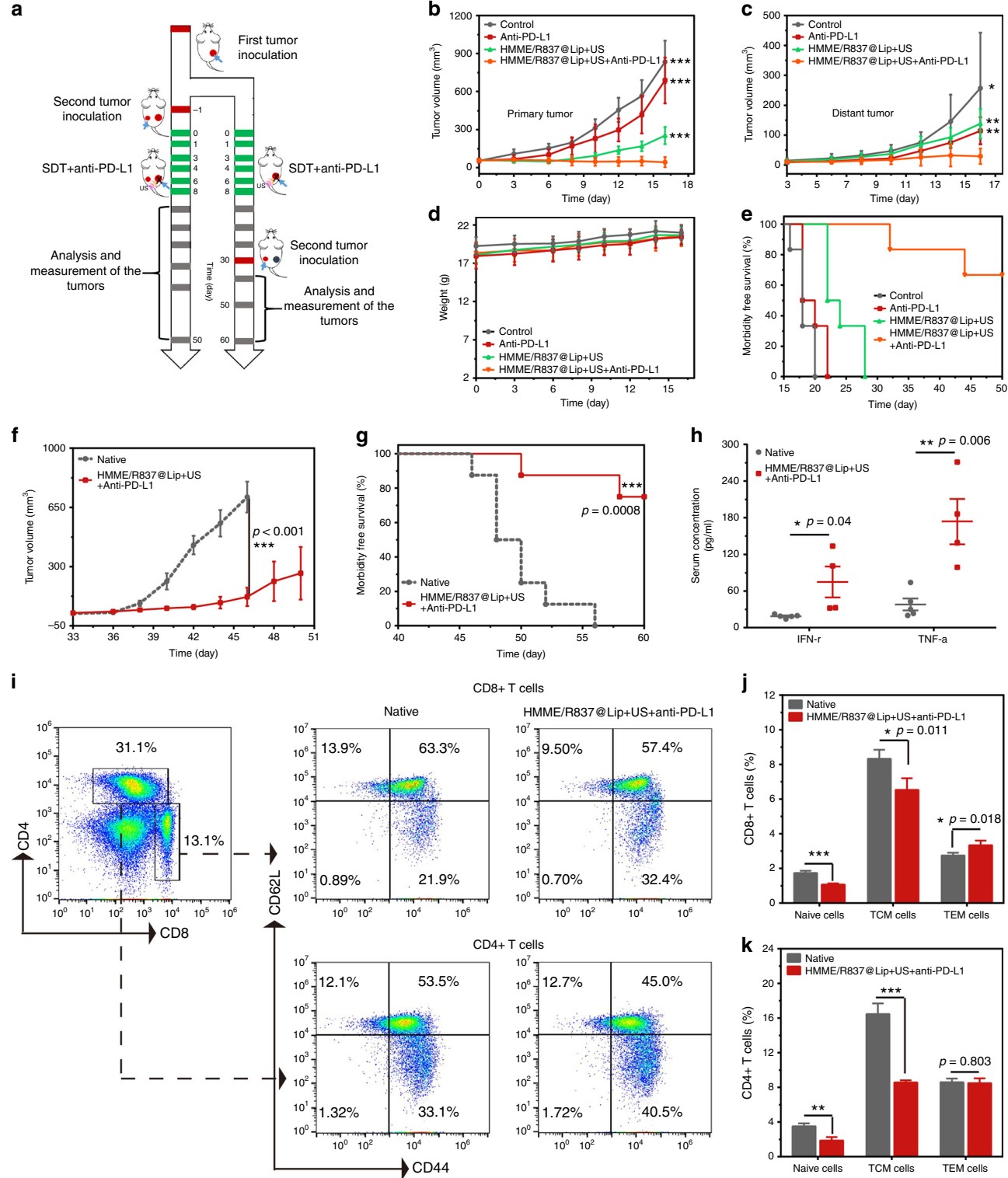

unresolved or disputed biosafety of previously reported inorganic nanosonosensitisers for enhanced SDT[29,48], all three components in HMME/IR837@Lip nanosonosensitisers have been clinically approved. Therefore, the designed nanosonosensitisers and the efficient combined therapeutic strategy are highly promising towards further clinical translation.

In summary, we report on the rational combination of nanosonosensitiser-augmented SDT and anti-PD-L1 checkpoint blockade-enabled immunotherapy for highly efficient tumour therapy, which is based on the construction of a multifunctional

nanosonosensitiser (HMME/R837@Lip) with the encapsulated sonosensitisers and immune adjuvant. After systemic administration, HMME/R837@Lip nanosonosensitisers acquire high tumour accumulation and long-time tumour retention, thereby effectively inducing tumour-cells death upon implementing repeated noninvasive US irradiations. By employing two tumour types and various tumour models, we have demonstrated that HMME/R837@Lip-augmented SDT in combination with anti-PD-L1 not only efficiently suppressed the primary tumour growth but also substantially prevented the mimic distant and lung

**Fig. 8** Anticancer activity of HMME/R837@Lip-augmented SDT plus anti-PD-L1 therapy in murine colorectal cancer models. **a** Schematic illustration of the experiment design to assess the antitumour immune responses against mimic distant tumours and the immunological memory response triggered by HMME/R837@Lip-augmented SDT and anti-PD-L1 combination therapy; **b**, **c** primary (**b**) and mimic distant (**c**) tumour growth curves of different groups of tumour-bearing mice after various treatments as indicated in the figure. Error bars are based on SD ($n = 6$); **d** time-dependent body-weight surveillance of mice ($n = 6$) after different treatments; **e** morbidity-free survival of mice after different treatments ($n = 6$); **f** tumour-growth curves of the rechallenged tumours in the corresponding groups were stopped when the first mouse died. Error bars are based on SE ($n = 8$); **g** morbidity-free survival of mice after the indicated treatment ($n = 8$), statistical significance was calculated via the log-rank (Mantel–Cox) test; **h** TNF-α and IFN-γ levels in serum isolated from mice of the treatment group and native group 7 days after the second tumour was introduced. Error bars are based on SE ($n = 5$ for the native group, $n = 4$ for the combined treatment group); **i** representative flow-cytometry plots of splenic lymphocytes of CT26 tumour-bearing mice treated with the combined immunotherapy at day 29 right before mice rechallenged with the secondary tumours (gating on CD3$^+$ cells); **j**, **k** absolute quantification of naive T cells, TCM and TEM in the spleen. TCM central memory T cells, TEM effector memory T cells. Error bars are based on SD ($n = 4$). Statistical significances were calculated via Student's $t$ test. *$p < 0.05$, **$p < 0.01$ and ***$p < 0.001$

metastasis. Systematic immune responses, including elevation of CD45$^+$ leucocytes and CD8$^+$ and CD4$^+$ Teffs, as well as the enhanced DC maturation and cytokine secretion, have been shown to be responsible for the enhanced immunotherapy and inhibited metastasis. Furthermore, our combined tumour immunotherapy strategy also offers a long-term immunological memory function, which can protect against tumour rechallenge after elimination of initial tumours. Considering that SDT-based cancer therapy has already been used in the clinic and the main components within the designed nanosonosensitisers are all FDA-approved agents, this may be a promising therapeutic modality for treating cancer.

## Methods

**Materials**. Dipalmitoyl phosphatidylcholine (DPPC), 1,2-distearoyl-sn-glycero-3-phosphoethanolamine-N-[amino(polyethylene glycol)2000] (DSPE-PEG-2000), Cy5.5-DSPE-PEG and cholesterol were purchased from Xi'an Ruixi Biological Technology Co., Ltd. Imiquimod (R837) was purchased from Sigma-Aldrich. Hematoporphyrin monomethyl ether (HMME) was provided by Di-Bai Chemical Technology Co., Ltd. Dimethyl sulfoxide (DMSO), methanol and trichloromethane (CHCl$_3$) were purchased from the Sinopharm Chemical Reagent Co. Anti-PD-L1 was obtained from Bioxcell (α-PD-L1, Clone: 10 F.9G2, Catalog No. BE0101). ELISA kit was purchased from R&D Systems, USA.

**Preparation of HMME/R837@Lip nanosonosensitisers**. HMME/R837@Lip was synthesised via a typical reverse evaporation method[20,27]. Briefly, R837 and HMME were firstly dissolved in DMSO (2.5 mg/ml) and methanol (6 mg/ml), respectively. Afterwards, R837 (0.24 ml) and HMME (0.1 ml) solution were in sequence added into the trichloromethane solution containing DPPC, DSPE-PEG-2000 and cholesterol with a fixed weight ratio 3:1:1. The following evaporation on a rotary evaporator at 100 mbar and 100 rpm at 60 °C for 1 h was carried out. After that, the pressure was decreased to 0 mbar, and the mixture was rotated and evaporated overnight, aiming to completely remove the solvent and obtain the lipid thin film. Then, 5 ml of PBS (0.01 M, pH = 7.6) was added into the lipid film, and the mixture was rotated at 100 rpm at 55 °C in an oil bath for another 1 h. Ultimately, the vesicles were harvested via an extrusion process by mini-extruders using a 200- and 100 -nm membrane, respectively, and then were further purified by dialysis.

**Nanosonosensitiser characterisation**. The characterisations of the morphology and structure of HMME/R837@Lip were conducted on TEM using a JEM-2100F electron microscope. DLS was employed to measure the hydrodynamic particle size and zeta potential using the Malvern Zetasizer Nanoseries (Nano ZS90). The encapsulation efficiency of HMME in the nanosonosensitisers was evaluated by UV-vis spectra technology (UV-3101PC Shimadzu spectroscope) and HPLC. The R837 encapsulated in the nanosonosensitisers was determined by HPLC (Agilent 1260).

The $^1O_2$ generation from HMME/R837@Lip upon exposure to US irradiation was detected by TEMP (Dojindo Molecular Technologies, Inc.). Typically, HMME/R837@Lip (150 μg/ml) was treated by US irradiation (1.0 MHz, 1.5 W/cm$^2$, 50% duty cycle) for 1 min in the presence of TEMP (97 μM). Immediately afterwards, the $^1O_2$ generation was detected by the electron paramagnetic resonance (ESR) spectrometer. As control groups, HMME/R837@Lip + TEMP and US + TEMP were also tested for comparison.

As for the quantitative analysis of $^1O_2$ generation, HMME/R837@Lip was suspended in PBS (150 μg/ml), followed by adding DPBF (Sigma-Aldrich, 40 μL, 8 mM). Then, the mixture was exposed to US irradiation (1.0 MHz, 1.5 W/cm$^2$, 50% duty cycle) every 1 min, and then the absorbance intensity at the wavelength of 398 nm was recorded using a UV-vis spectroscope.

**Cell experiment**. The 4T1 murine breast-cancer cell line and CT26 murine colorectal cancer-cell line were originally obtained from American Type Culture Collection (ATCC) and cultured in the recommended medium and condition. FLuc-4T1 cancer-cell line was purchased from Hanyin Biotechnology Co., Ltd.

**Cellular uptake and efflux**. The 4T1 cancer cells seeded in six-well plates ($2 \times 10^5$ cells per well) were incubated with HMME/R837@Lip (100 μg/mL) for 1, 2, 4 and 24 h, respectively. Then, the treated cells were collected, washed three times with PBS, counted on a haemocytometer and finally lysed with 0.5% (w/v) sodium dodecyl sulfate (SDS, pH 8.0) to release uptaken HMME and quantitatively test the concentration using UV-vis according to the standard curve of concentration-dependent UV-Vis absorbance intensity. Thus, the uptake level of particles was represented by the amount of HMME (μg) per $10^6$ cells.

To investigate the efflux of HMME, 4T1 cells were incubated with HMME/R837@Lip (100 μg/ml) for 4 h. Then, the culture medium was discarded and the adherent cells were washed with PBS three times. Subsequently, 2 ml of fresh culture medium was added to each well and another incubation for 1, 2, 4 and 24 h was conducted. The culture media was collected to determine the HMME concentration via a UV-Vis method so as to quantify HMME efflux after adding 0.5% Triton X-100 into the medium. The results were expressed as the percent of the amount of HMME being effluxed compared with the 4-h cellular uptake amount.

The intracellular distribution of HMME/R837@Lip nanosonosensitisers was also directly observed on CLSM. The 4T1 cancer cells were incubated with HMME/R837@Lip nanosonosensitisers for 1, 2, 4 and 24 h, respectively, which were then washed with PBS three times, fixed with 4% paraformaldehyde and observed under CLSM using a 405 -nm laser source.

**In vitro $^1O_2$ generation at the cellular level**. Six groups (i.e., control without any treatment, HMME@Lip, HMME/R837@Lip, US, HMME@Lip + US and HMME/R837@Lip + US) were set to explore the potential of HMME/R837@Lip nanosonosensitisers in producing $^1O_2$ upon exposure to US irradiation. In brief, 4T1 cells or CT26 cells were incubated with HMME/R837@Lip (100 μg/ml in DMEM) at 37 °C for 4 h. Then, the culture media was replaced by DCFH-DA (100 μl, 1/9 μl in DMEM, Beyotime Biotechnology) and US irradiation (1.0 MHz, 1.5 W/cm$^2$, 50% duty cycle, 1 min) was implemented. After incubating for another 30 min, the cells were washed with PBS three times and then observed by CLSM. In other comparison groups, the differences lied in the replacement of HMME/R837@Lip by HMME@Lip with exposure to US and US-free irradiation and the other procedures were identical.

**In vitro cytotoxicity assay**. The evaluation of cell apoptosis after SDT was enforced by CLSM. 4T1 cells or CT26 cells were incubated with HMME@Lip, or HMME/R837@Lip at a HMME concentration of 100 μg/ml for 24 h, was treated as mentioned above and then incubated for another 24 h. After that, the cells were stained by Calcein-AM/PI, followed by analysis using CLSM.

The sonotoxicity against cancer cells was further evaluated using CCK-8 assay. 4T1 cells were seeded in 96-well plates ($3 \times 10^3$ cells per well) and incubated with HMME@Lip, or HMME/R837@Lip at different HMME concentrations for 24 h. Subsequently, they were exposed to US irradiation (1.0 MHz, 1.5 W/cm$^2$, 50% duty cycle, 1 min) and then incubated for another 24 h. Cell viability was evaluated by CCK-8 assay based on the absorbance at the wavelength of 450 nm by using a microplate reader (Bio-TekELx800, USA).

**In vitro DC stimulation transwell experiment**. Bone-marrow-derived DCs were isolated from 8-week-old BALB/c mice according to an established method[52]. Firstly, residual 4T1 cancer cells after different treatments were put in the upper compartment of the transwell system, while the DCs were seeded in the lower compartment. Lipopolysaccharide (LPS, Sigma) at a dose of 1 μg/ml was used as the positive control. After various treatments, DCs stained with anti-CD11c FITC, anti-CD80 APC and anti-CD86 PE were analysed by FCM (BD LSRFortessa). The

proinflammatory cytokines (i.e., IL-6 and TNF-α) from DC suspension were tested by using ELISA kits with a standard protocol.

**HMME/R837@Lip-augmented SDT for immune system activation**. Female BALB/c mice (6–8 weeks) were purchased from Bikai Biological Technology Co. Ltd. All in vivo experiments were performed according to protocols approved by the Laboratory Animal Center of Shanghai Tenth Peoples' Hospital and were in accordance with the policies of the National Ministry of Health. Mice were randomly divided into six groups ($n = 3$), including: (1) control, (2) HMME @Lip, (3) HMME/R837@Lip, (4) US alone, (5) HMME@Lip + US and (6) HMME/R837@Lip + US. 4T1 cells ($1 \times 10^6$) were subcutaneously injected into the right chest of each mouse. Six days later, the tumours were allowed to reach ~70 mm$^3$ before the experiments. PBS was intravenously injected into the control and US group. The mice in other groups were administered with nanosonosensitisers intravenously at the HMME dose of 8 mg/kg and R837 dose of 6 mg/kg (150 μl). In the US irradiation groups (i.e., US alone, HMME@Lip + US and HMME/R837@Lip + US), US irradiation (1.0 MHz, 1.5 W/cm$^2$, 50% duty cycle, 5 min) was conducted 12 h post injection. On the third day, tumours were collected, dissected and subjected to H&E staining, TUNEL assay or CRT immunofluorescence staining. The tumour-draining lymph nodes were excised for analysis by FCM after co-staining with anti-CD11c FITC, anti-CD80 APC and anti-CD86 PE. Meanwhile, the blood samples were collected at 24, 48 and 72 h after treatments, and the proinflammatory cytokines, including IL-6, TNF-α and IL-12 p70 serum, were tested by ELISA.

**SDT plus PD-L1 blockade for suppression of distant tumours**. Objective to the first tumour inoculation, 4T1 cells ($1 \times 10^6$) or CT26 cells ($1 \times 10^6$) were subcutaneously injected into the right chest/flank of each mouse. Six days later, the tumour-bearing mice were divided into four groups ($n = 5$–6) randomly, including: (1) PBS, (2) anti-PD-L1, (3) HMME/R837@Lip + US and (4) HMME/R837@Lip + US + anti-PD-L1. 4T1 cells or CT26 cells were subcutaneously injected into the left chest/flank of each mouse for the second tumour inoculation. HMME/R837@Lip was i.v. injected into animals at the same doses with that mentioned above on days 0 and 3. US irradiations that shared the identical parameters with those in the above experiments were performed after 12 and 24 h post injection. Anti-PD-L1 antibodies at the dose of 75 μg/mouse were administered on days 1, 4, 6 and 8. The tumour volume was calculated according to the following formula: (width$^2 \times$ length)/2. At the end of the experiment, mice were killed and tumours were excised, weighed and photographed. Based on the standard animal protocol, the mice with tumours exceeding 1000 mm$^3$ would be euthanised in this work.

To establish a 4T1 orthotopic murine breast-cancer model, 4T1 cells ($5 \times 10^4$) were injected into the mammary fat pads of mice on the right side of the abdomen. Six days later, 4T1 cells with the same number of cells were injected into the left mammary fat pads of the mice to establish the bilateral tumour model. To establish the lung metastasis model, fLuc-4T1 cells ($1 \times 10^5$) were administered intravenously via tail vein infusion. The followed treatments were the same as the aforementioned procedures.

An in vivo imaging spectrum system was used for the bioluminescence imaging of mice after 60 s of exposure. At the end of this experiment, lungs were collected and fixed in Bouin's solution. Lung micrometastases in five lobes were counted directly through microscopic observation, and afterwards they were studied by pathological analysis.

**Mechanism investigation of the in vivo combined therapeutics**. To systematically investigate the in vivo antitumour immune responses against mimic distant tumours, the tumours were harvested and treated with the tissue dissociation kit (Miltenyi Biotec, Germany) to produce a single-cell suspension according to the specified procedures. The harvested cells were further stained with several fluorochrome-conjugated antibodies: CD45-FITC (BD, Catalog: 561088), CD3-PerCP-Cy5.5 (BD, Catalog: 551163), CD4-BV510 (BD, Catalog: 563106), CD8-BV421 (BD, Catalog: 563898), NKp46-APC (Biolegend, Catalog: 137608), B220-PE-Cy7 (BD, Catalog: 552772) and Foxp3-PE (eBioscience, Catalog: 12–4771) and then analysed by FCM. For the analysis of the memory T cells, spleen cells of mice were harvested and stained with anti-CD3-FITC (Biolegend, Catalog: 100306), anti-CD4-PE-Cy7 (eBioscience, Catalog: 25–0041), anti-CD8-PerCP-Cy5.5 (eBioscience, Catalog: 45–0081), anti-CD44-PE (eBioscience, Catalog: 12–0441) and anti-CD62L-APC (eBioscience, Catalog: 17–0621) antibodies and then analysed by FCM. All antibodies were diluted ~100 times.

**Statistical analysis**. Data were expressed as means ± standard deviation (SD)/standard error (SE) and were compared by means of an unpaired Student's $t$ test or Mann–Whitney U test. All the statistical analyses were conducted using the SPSS software (version 18.0).

## Data availability

All data are either included within the paper and Supplementary Information file or available from the authors upon reasonable request.

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

## Acknowledgements

This work was supported in part by the National Key R&D Program of China (Grant no. 2016YFA0203700), the Science and Technology Commission of Shanghai Municipality (Grants 14441900900, 16411971100) and the National Natural Science Foundation of China (Grants 81725008, 81771836, 81671695, 81601501, 81601502, 81501473, 81501474, 51722211 and 51672303) and Program of Shanghai Subject Chief Scientist (Grant no. 18XD1404300).

## Author contributions

Y.C. and W.Y. conceived this project. Y.C., H.X., W.Y. and K.Z. designed and supervised the project and commented on the project. W.Y., L.C., L.Y., B.Z., H.Y., W.R., C.L., L.G., Y.Z. and L.S. synthesised and characterised the nanosonosensitisers, performed in vitro and in vivo experiments and analysed the data. W.Y. wrote the paper. All the authors contributed to the discussion during the whole project.

## Additional information

**Competing interests:** The authors declare no competing interests.

