## [Peer Review File · Nature Communications]

Reviewers' Comments:

Reviewer #1:

Remarks to the Author:

In this manuscript, the authors reported the utilization of liposomes encapsulated with sonosensitizers (HMME) and immune adjuvant (R837) for combined sonodynamic therapy and anti-PD-L1 based immune therapy. It was uncovered that such combination therapy could not only efficiently suppress the primary tumor growth, but also prevent the growth of those mimic distant tumor and lung metastasis. However, similar combination treatment strategy has been previously reported in Ref. 16 & 38, which demonstrated that the combination utilization of R837 with photothermal therapy or photodynamic therapy could prime strong specific immune attack on those metastatic tumors. Therefore, I think that the scientific novelty of this work is limited though this is a complementary research work.

Some comments:

1. I think that it may not be appropriate to administrate R837 via intravenous injection route since such strategy is prone to induce severe side effect (e.g. cytokine storms).
2. I would like to suggest the authors to confirm if such combination treatment would work well on other tumor models.
3. I want to know if such treatment could induce effective long-term immune-memory effects.
4. For intracellular ROS detection (Figure 3a), it seems unreasonable that the whole 4T1 cell was stained by DCFH-DA, including nucleus. Please carefully check it.
5. In Figure 3d&e, why HMME@Lip treated group (without US irradiation) could also promote DC maturation? Please explain it.
6. It was unreasonable to perform the sonodynamic therapy (Figure 4) before evaluating the in vivo behaviors of HMME/R837@Lip (Figure 5a-d).

Reviewer #2:

Remarks to the Author:

In this manuscript Yue et al described a highly efficient combined tumour-therapeutic modality based on the rational integration of nanosonosensitizers-augmented noninvasive sonodynamic therapy (SDT) and checkpoint-blockade immunotherapy. The constructed HMME/R837@Lip nanosonosensitizers produce toxic singlet oxygen under repeated noninvasive ultrasound irradiations to destroy the primary tumours, whose debris acts as tumour-associated antigens, together with R837 as the adjuvant to trigger strong antitumour immune responses. This combination therapy not only efficiently arrests primary tumour progression, but also substantially prevents tumour metastasis on the subcutaneous, orthotopic as well as artificial whole-body metastasis tumour models. Overall, it is a piece of excellent work. I suggest acceptance of this manuscript after addressing following concerns.

- 1) The combination treatment induced strikingly high and sustained levels of IL-12, IL-6, and TNF- α in serum, which the authors demonstrated to indicate potency, but systemic cytokines in patients lead to toxicity- systemic IL-12 could be lethal in humans. The authors should present some analysis of safety of the treatment- measurement of animal body weights, histopathology of organs, serum analysis for liver enzymes indicative of liver damage, to evaluate the level of toxicity of the therapy.
- 2) The T cell infiltration analyses in figure 7d should be reported as raw cell counts per mass of tumor, as percentages of cells can be misleading since the frequencies of many cell types in the tumors may be changing simultaneously following treatment.
- 3) While the findings are interesting and extensive on 4T1 tumor model, other tumor cell lines may be used.

4) The authors should comment on the potential clinical translation of their strategy, in particular the use of laser-based treatment of cancer.

5) Latest studies/reviews in this field are suggested to cite, e.g., Nature Biomedical Engineering 1 (2), 0011, 2017; Nature Nanotechnology volume 12, 877–882, 2017; Sci. Immunol., 2017, 2(eaan5692).

Response to Reviewer 1

Comments and suggestions from Reviewer 1.

In this manuscript, the authors reported the utilization of liposomes encapsulated with sonosensitizers (HMME) and immune adjuvant (R837) for combined sonodynamic therapy and anti-PD-L1 based immune therapy. It was uncovered that such combination therapy could not only efficiently suppress the primary tumor growth, but also prevent the growth of those mimic distant tumor and lung metastasis. However, similar combination treatment strategy has been previously reported in Ref. 16 & 38, which demonstrated that the combination utilization of R837 with photothermal therapy or photodynamic therapy could prime strong specific immune attack on those metastatic tumors. Therefore, I think that the scientific novelty of this work is limited though this is a complementary research work.

Response: Thank you very much for the kind comment and suggestion. As the reviewer mentioned, the combined utilization of R837 with photothermal therapy or photodynamic therapy has been demonstrated. However, the traditional photo-triggered therapies suffer from the low tissue-penetrating depth of light, hindering their further application in tumor therapy.

Herein, we report, for the first time, on a highly efficient combined tumour-therapeutic modality based on the rational integration of nanosonosensitizers-augmented noninvasive sonodynamic therapy (SDT) and checkpoint-blockade immunotherapy based the construction of a multifunctional nanosonosensitizers with all FDA-approved agents. The application of ultrasound as the triggering source for SDT could potentially solve the critical issue of light as the triggering source based on the fact that ultrasound is featured with high tissue-penetrating capability. Based on two tumour types (4T1 and CT26) and the subcutaneous, orthotopic, as well as artificial whole-body metastasis tumour models, it has been successfully demonstrated that the combined tumour treatment between nanosonosensitizers-augmented SDT and anti-PD-L1 antibody facilitates tumour-specific cytotoxic T-cell penetration into tumours and provokes vigorous anti-tumour immunity, which not only efficiently arrests primary tumour progression, but also substantially prevents the mimic distant and lung metastasis. Furthermore, such a

combined tumour immunotherapy strategy offers a long-term immunological memory function, which can protect against tumour rechallenge after elimination of the initial tumours. Therefore, this work represents the proof-of-concept combinatorial tumour therapeutics based on noninvasive tumours-therapeutic modality integrated with immunotherapy.

The specific originality and novelty of this work are clarified as follows.

1. Conceptual advance. We have, for the first time, proposed, established and demonstrated the proof-of-concept combinatorial tumour therapeutics based on the noninvasive tumour-therapeutic modality, *i.e.*, sonodynamic therapy and immunotherapy, making it highly promising to substitute traditional invasive radiofrequency ablation, ionizing radiation therapy, toxic chemotherapy or photo-therapies with low tissue-penetrating depth.
2. First paradigm of combination of checkpoint blockade with sonodynamic therapy. Ultrasound irradiation in SDT is much safer, and more preferable and applicable than other physical irradiations such as light irradiation in photo-therapies (*e.g.*, photothermal therapy or photodynamic therapy), because ultrasound as the safe irradiation source can acquire a much higher tissue-penetrating depth than light, enabling the treatment of deep-seated tumours. Successfully combined antitumour-treatment results have been demonstrated in this systematic work by using various types of tumor models, thus it holds considerable potential as a next-generation anti-tumor strategy not only for treating superficial tumours but also for combating deep-seated tumours.
3. Highly promising towards further clinical translation. Compared to unresolved or disputed biosafety issues of previously developed nanosonosensitizers for enhanced SDT, all the main components in the constructed nanosonosensitizers in this work have been approved for clinical use by FDA. Therefore, the designed nanosonosensitizers and their corresponding efficient combined therapeutic protocol are highly promising towards further clinical translation and application.

Please find the following detailed responses to your comments and suggestions.

(1) I think that it may not be appropriate to administrate R837 *via* intravenous injection route since such strategy is prone to induce severe side effect (e.g. cytokine storms).

Response: Thank you very much for the constructive suggestion. We agree with the reviewer's opinion that administration of R837 *via* intravenous injection route would possibly induce severe side effects (e.g. cytokine storms). Therefore, before R837 was employed in this work, we had made a sufficient and systematic literature review, including its delivery route, safety and efficacy. In addition, other immune regulators which have a similar biological property as R837 have been discussed.

Chen et al.¹ used mPEG-PLGA to encapsulate both ICG and R837, which showed high accumulation in tumours upon *i.v.* injection. Also, all mice behaved normally after the treatment by *i.v.* injection, without significant body-weight fluctuation or accidental death. In addition, Rodell et al.² and Liu et al.³ designed multifunctional nanoparticle systems encapsulating R848, another anti-tumour immune regulator with a similar biological property as R837. Their results did not show seriously systemic toxicity after intravenous injection. These findings evinced that these immune regulators encapsulated by nanoparticle system could potentially be used by intravenous-injection route.

In this work, in order to explore the potential harmful effect of the intensified immune therapeutic effect, physical conditions of mice including serum biochemistry assay, body temperature and body weight were carefully monitored during the entire study. It was recorded that all the measured serum biochemistry indexes of the treated groups were not significantly different from those of the healthy control (**Supplementary Fig. 7**), indicating that such a boosted anti-tumour immunity was tolerable by mice. No abnormal body-temperature or body-weight changes were observed in mice (**Supplementary Fig. 9a-b**), and no evident damages to the major organs (*i.e.*, heart, liver, lung, spleen and kidney) in pathological analysis also confirmed the high histocompatibility (**Supplementary Fig. 8**).

For the administration of R837 *via* intravenous injection route, we think that these desirable results might be attributed to two aspects. On one hand, we used a nanoparticle system (liposome) as the carrier for encapsulating R837, which could efficiently acquire high accumulation into

tumours and lower retention in other organs after *i.v.* administration, as contributed by the long blood-circulation duration and high stability in favor of enhanced permeability and retention (EPR) effect. On the other hand, a dose of 6 mg/kg R837 was used in this work, and this was much lower than the dose of anti-tumour recommended by oral route (50 mg/kg)⁴. Therefore, R837 could be administrated *via* intravenous injection route into mice if it was delivered by an efficient nanocarrier at a proper dose. We have clarified this issue in the revised manuscript which is marked in yellow (Page 10).

References:

1. Chen, Q. et al. Photothermal therapy with immune-adjuvant nanoparticles together with checkpoint blockade for effective cancer immunotherapy. *Nat Commun* **7**, 13193 (2016).
2. Rodell, C.B. et al. TLR7/8-agonist-loaded nanoparticles promote the polarization of tumour-associated macrophages to enhance cancer immunotherapy. *Nat Biomed Eng* **2**, 578-588 (2018).
3. Liu, Y. et al. Dual pH-responsive multifunctional nanoparticles for targeted treatment of breast cancer by combining immunotherapy and chemotherapy. *Acta Biomater* **66**, 310-324 (2018).
4. Miller, R.L. et al. Imiquimod applied topically: a novel immune response modifier and new class of drug. *Int J Immunopharmacol* **21**, 1-14 (1999).

(2) I would like to suggest the authors to confirm if such combination treatment would work well on other tumor models.

Response: Thank you very much for the constructive suggestion. According to the reviewer's suggestion, we further explored whether such a combined tumour immunotherapy could potentially be used for other tumour types by further employing murine colorectal CT26 cancer cell as the model in addition to the breast 4T1 cancer model. Both *in vitro* and *in vivo* systematic evaluations have been further conducted. It has been found that HMME/R837@Lip+US also

exhibited remarkable sonotoxicity against CT26 cancer cells (**Supplementary Fig. 12**), and such a combination treatment could significantly inhibit the primary and mimic distant tumours (**Fig. 8b, c, Supplementary Fig. 14**). We also found that four out of six mice survived for 50 days after such a combined tumour immunotherapy, as compared with mice in the other three control groups that all died within 16–28 days (**Fig. 8e**), demonstrating that the survival rate of tumor-bearing mice could be significantly improved by this combined tumour immunotherapy. In addition, the survived four mice in the combined treatment group were sacrificed at day 50 for the careful necropsy, and no visually noticeable metastatic tumours were discovered. Besides, all the mice behaved normally and no abnormal body-temperature (**Supplementary Fig. 13**) or body-weight changes (**Fig. 8d**) were observed during the entire study. These results successfully demonstrate that our findings on the conducted immunotherapy strategy in 4T1 cancer models can be extended to other types of tumours.

(3) I want to know if such treatment could induce effective long-term immune-memory effects.

Response: Thank you very much for the constructive suggestion. The immunological memory response is a well-known feature of the adaptive immunities. Therefore, according to the reviewer's suggestion, we further assessed the immune memory effects of our combined tumour immunotherapy strategy by rechallenging mice with the secondary inoculation of CT26 cancer cells on day 30, which was the 22th day after SDT plus anti-PD-L1 treatment to remove the first tumours. At the same time, eight age- and sex-matched native mice were inoculated with the same number of CT26 cells to help identify the results. It showed that, the growth of re-inoculated tumours in combined treatment group was obviously inhibited in compared with that of native group (**Fig. 8f, Supplementary Fig. 15**). And 75% (6/8) of the mice after combined treatment performed a remarkable resistance to rechallenge. Comparatively, all the control mice developed tumours and died within 26 days after inoculation (**Fig. 8g**). These results demonstrated that the excellent long-term immune memory effects were generated by HMME/R837@Lip-augmented SDT plus anti-PD-L1 blockade treatment.

(4) For intracellular ROS detection (Figure 3a), it seems unreasonable that the whole 4T1 cell was stained by DCFH-DA, including nucleus. Please carefully check it.

Response: We thank you for the kind reminding, which is highly appreciated. In order to check the results of CLSM images with 4T1 cells, in our further study during the revision round, we also explored the intracellular ROS production in CT26 cells using CLSM observation as stained by DCFH-DA. And the results of CT26 cells were the same as 4T1 cells (**Supplementary Fig. 11**). To further verify this result, we also conducted a careful related literature review. Han et al.¹ and Huang et al.² also showed that the cells with high ROS content displayed entirely in green on CLSM images if stained with DCFH-DA. These results might be related to the cellular structural features and angle of photographing. When the cells were flat in the plate, the nucleus would be surrounded with cytoplasm stained green under the confocal microscope. Therefore, if we photographed on top of the cellular level, the whole cell would be in green without specific staining for nucleus.

References:

1. Han, X. et al. Oxygen-deficient black titania for synergistic/enhanced sonodynamic and photoinduced cancer therapy at near infrared-II biowindow. *ACS Nano* **12**, 4545–4555 (2018).
2. Huang, P. et al. Metalloporphyrin-encapsulated biodegradable nanosystems for highly efficient magnetic resonance imaging-guided sonodynamic cancer therapy. *J Am Chem Soc* **139**, 1275-1284 (2017).

(5) In Figure 3d&e, why HMME@Lip treated group (without US irradiation) could also promote DC maturation? Please explain it.

Response: Thank you very much for the constructive question, which is highly appreciated. Many studies showed that some sensitizers¹ including HMME² had some dark toxicity against cancer cells, that is to say, even without US irradiation, some cancer cells co-cultured with HMME@Lip would partially die to some extent because of the dark toxicity. In this work, as

shown in **Supplementary Fig. 4a and Fig. 3b**, approximately 8% 4T1 cells died because of the dark toxicity of HMME (100 µg/mL). Therefore, if residual 4T1 cancer cells after the incubation with HMME@Lip were co-cultured with DC cells, they could slightly help the promotion of DC maturation. We have clarified this issue in the revised manuscript which is marked in yellow (Page 8).

References:

1. He, C. et al. Core-shell nanoscale coordination polymers combine chemotherapy and photodynamic therapy to potentiate checkpoint blockade cancer immunotherapy. *Nat Commun* **7**, 12499 (2016).
2. Liu, F. et al. Ultrasound-guided tumor sonodynamic therapy based on sonosensitizer liposome. *Chem. Lett* **45**, 1304-1306 (2016).

(6) It was unreasonable to perform the sonodynamic therapy (Figure 4) before evaluating the *in vivo* behaviors of HMME/R837@Lip (Figure 5a-d).

Response: We thank the reviewer very much for this kind comment, which is highly appreciated. Firstly, because the three main components within the designed nanosonosensitizers (HMME/R837@Lip) are all FDA-approved agents, and the doses of these components used in this study had already been systematically verified in previous studies. Therefore, the intrinsic toxicity of as-designed nanosonosensitizers is low. Secondly, in part **Fig. 4**, we conducted the serum biochemistry assay to evaluate the anti-tumour immune responses as well as the biological safety of HMME/R837@Lip-augmented SDT. These results showed that HMME/R837@Lip-augmented SDT only increased secretion of the pro-inflammatory cytokines within 72 h. We did not observe any notable cytokine-storm-like side effects (**Fig. 4f-h**). All the mice behaved normally after treatments with *i.v.* injected HMME/R837@Lip without accidental death. These results evidenced our further study for the combined HMME/R837@Lip-augmented SDT with anti-PD-L1 treatment. We have clarified this issue in the revised manuscript which is marked in yellow (Page 10). Thirdly, the safety evaluation in part **Fig. 5a-d** including abnormal temperature, body-weight and the damages to the major

organs (*i.e.*, heart, liver, lung, spleen and kidney) mainly focused on the combined treatment strategy.

Response to Reviewer 2

Comments and suggestions from Reviewer 2.

In this manuscript Yue et al. described a highly efficient combined tumour-therapeutic modality based on the rational integration of nanosonosensitizers-augmented noninvasive sonodynamic therapy (SDT) and checkpoint-blockade immunotherapy. The constructed HMME/R837@Lip nanosonosensitizers produce toxic singlet oxygen under repeated noninvasive ultrasound irradiations to destroy the primary tumours, whose debris acts as tumour-associated antigens, together with R837 as the adjuvant to trigger strong antitumour immune responses. This combination therapy not only efficiently arrests primary tumour progression, but also substantially prevents tumour metastasis on the subcutaneous, orthotopic as well as artificial whole-body metastasis tumour models. Overall, it is a piece of excellent work. I suggest acceptance of this manuscript after addressing following concerns.

Response: Thank you very much for the kind comment, suggestion and recommendation. Please find the following detailed responses.

(1) The combination treatment induced strikingly high and sustained levels of IL-12, IL-6, and TNF- α in serum, which the authors demonstrated to indicate potency, but systemic cytokines in patients lead to toxicity- systemic IL-12 could be lethal in humans. The authors should present some analysis of safety of the treatment-measurement of animal body weights, histopathology of organs, serum analysis for liver enzymes indicative of liver damage, to evaluate the level of toxicity of the therapy.

Response: Thank you for the kind reminding, which is highly appreciated. In this work, we have conducted a systematic biosafety evaluation including serum biochemistry assay, body temperature, body weight as well as histopathology of organs. It was recorded that all the measured serum biochemistry indexes of the treated groups were not significantly different from those of the healthy control (**Supplementary Fig. 7**), indicating that such a boosted anti-tumour immunity was tolerable by mice. No abnormal temperature and evident body-weight changes

(**Supplementary Fig. 9**) were observed in mice, and no evident damages to the major organs (*i.e.*, heart, liver, lung, spleen and kidney) in pathological analysis (**Supplementary Fig. 8**). We have clarified this issue in the revised manuscript which is marked in yellow (Page 10, 15, 18).

(2) The T cell infiltration analyses in Fig. 7d should be reported as raw cell counts per mass of tumor, as percentages of cells can be misleading since the frequencies of many cell types in the tumors may be changing simultaneously following treatment.

Response: Thank you very much for the constructive suggestion. According to the reviewer's suggestion, absolute quantification of the T cells in the tumours has been displayed in the revised manuscript (**Fig. 7d**). Also, the significance of the data was re-analyzed according to a Student's t test (**Fig. 7d, e**).

(3) While the findings are interesting and extensive on 4T1 tumor model, other tumor cell lines may be used.

Response: Thank you very much for the constructive suggestion. According to the reviewer's suggestion, we further explored whether such a combined tumour immunotherapy could potentially be used for other tumour types by further employing murine colorectal CT26 cancer cell as the model in addition to the breast 4T1 cancer model. Both *in vitro* and *in vivo* systematic evaluations have been further conducted. It has been found that HMME/R837@Lip+US also exhibited remarkable sonotoxicity against CT26 cancer cells (**Supplementary Fig. 12**), and such a combination treatment could significantly inhibit the primary and mimic distant tumours (**Fig. 8b, c, Supplementary Fig. 14**). We also found that four out of six mice survived for 50 days after such a combined tumour immunotherapy, as compared with mice in the other three control groups that all died within 16–28 days (**Fig. 8e**), demonstrating that the survival rate of tumor-bearing mice could be significantly improved by this combined tumour immunotherapy. In addition, the survived four mice in the combined treatment group were sacrificed at day 50 for the careful necropsy, and no visually noticeable metastatic tumours were discovered. Besides, all the mice behaved normally and no abnormal body-temperature (**Supplementary Fig. 13**) or

body-weight changes (**Fig. 8d**) were observed during the entire study. These results successfully demonstrate that our findings on the conducted immunotherapy strategy in 4T1 cancer models can be extended to other types of tumours.

(4) The authors should comment on the potential clinical translation of their strategy, in particular the use of laser-based treatment of cancer.

Response: Thank you for your kind suggestion. Considering that SDT-based cancer therapy has already been used in clinic and the main components within the designed nanosensitizers are all FDA-approved agents, it is highly encouraging and expected to find the further clinical translation of this therapeutic modality on combating cancer. As compared to traditional laser-triggered photonic therapy, SDT-based cancer therapy is expected to be more promising in clinic based on the high tissue-penetrating capability of ultrasound as the triggering source. We have clarified them in the revised manuscript which is marked in yellow (Page 22).

(5) Latest studies/reviews in this field are suggested to cite, *e.g.*, Nature Biomedical Engineering 1 (2), 0011, 2017; Nature Nanotechnology volume 12, 877–882, 2017; Sci. Immunol., 2017, 2(eaan5692).

Response: Thank you very much for the kind suggestion. According to the reviewer's suggestion, these latest studies have been cited in the revised manuscript (Ref. 10, 17, 18).

Reviewers' Comments:

Reviewer #1:

Remarks to the Author:

In this revised manuscript of "Nanosensitizer-Augmented Noninvasive Cancer Sonodynamic/Immuno-Therapy", the authors have carefully addressed the issues and comments raised during the last round revision. The quantity of this manuscript has been improved. However, some other revisions are still needed before this manuscript could be considered for acceptance.

1. It was reported that the quantity of released damage-associated molecular patterns (DAMPs, e.g. CRT, HMGB1, and ATP) from the dying cancer cells are positively correlated with the maturation of dendritic cells and subsequent activation of T cells for cancer immunotherapy. Therefore, I would like to strongly suggest the authors to evaluate the release profiles of these DAMPs from the tumors after such sono-dynamic treatment.

2. The cartoon picture of the HMME/R837@Lip in figure 1 is a somewhat misleading because it is hard to understand what the inner green part stands for.

3. The morphology of 4T1 cells in Figure 3a was shown in abnormal round feature, which was quietly different from that shown in Figure 3b.

4. The quality of those present figures is far from satisfactory, such as the font size of Figure 4d, 6, 7 and 8 is too small; the numeric scale of Figure 3 d is missing; the photograph of mice shown in Figure 5b and 6b&j is abnormally compressed.

Reviewer #2:

Remarks to the Author:

The authors have addressed my comments.

Response to Reviewer 1

Comments and suggestions from Reviewer 1.

In this revised manuscript of “Nanosonosensitizer-Augmented Noninvasive Cancer Sonodynamic/Immuno-Therapy”, the authors have carefully addressed the issues and comments raised during the last round revision. The quantity of this manuscript has been improved. However, some other revisions are still needed before this manuscript could be considered for acceptance.

Response: Thank you very much for the kind suggestion and recommendation. Please find the following detailed responses.

1. It was reported that the quantity of released damage-associated molecular patterns (DAMPs, e.g. CRT, HMGB1, and ATP) from the dying cancer cells are positively correlated with the maturation of dendritic cells and subsequent activation of T cells for cancer immunotherapy. Therefore, I would like to strongly suggest the authors to evaluate the release profiles of these DAMPs from the tumors after such sono-dynamic treatment.

Response: Thank you very much for the constructive suggestion, which is highly appreciated. We do agree with the reviewer’s opinion that the release profiles of some damage-associated molecular patterns (DAMPs) from the tumors after sonodynamic treatment should be evaluated. As a very important part of DAMPs, calreticulin (CRT) exposed on the cell surface is a distinct biomarker of immunogenic cell death (ICD) (*Annu Rev Immunol* 31, 2013). Once on tumor cell surface, CRT acts as an “eat me” signal, stimulating macrophages and DCs to engulf the dying cells and their apoptotic debris (*Immunity* 38, 2013). Therefore, in this study we assessed the levels of CRT exposure on the 4T1 tumours treated with HMME/R837@Lip plus irradiation. As shown in Supplementary Fig.5, HMME/R837@Lip-augmented SDT treatment significantly induced CRT expression, which was in line with the results of previous studies (*J Am Chem Soc* 138, 2016; *Cancer Sci* 109, 2018). We have clarified this issue in the revised manuscript

according to the reviewer's constructive suggestion, which is marked in yellow (Page 9, 10 and Supplementary Fig.5).

2. The cartoon picture of the HMME/R837@Lip in figure 1 is a somewhat misleading because it is hard to understand what the inner green part stands for.

Response: We thank you very much for the kind reminding, which is highly appreciated. The cartoon picture of the HMME/R837@Lip in both Figure 1 and Figure 2 have been updated in the revised manuscript to make them clearer and more understandable according to the reviewer's reminding (Page 36, 37).

3. The morphology of 4T1 cells in Figure 3a was shown in abnormal round feature, which was quietly different from that shown in Figure 3b.

Response: Thank you very much for the kind comment. As a matter of fact, the dying cells usually show the abnormal round feature compared with the living cells. Figure 3a shows the CLSM images of 4T1 cells stained with DCFH-DA that could specifically target and label ROS. Because both HMME@Lip and HMME/R837@Lip could intracellularly generate ROS under US irradiation to induce toxic effects and further cause cell apoptosis and/or necrosis, thus if we photographed the cancer cell after treatments, the whole cell would display the abnormal round feature because of the poor condition of the cell.

4. The quality of those present figures is far from satisfactory, such as the font size of Figure 4d, 6, 7 and 8 is too small; the numeric scale of Figure 3 d is missing; the photograph of mice shown in Figure 5b and 6b&j is abnormally compressed.

Response: Thank you very much for the constructive reminding, which is highly appreciated. The quality of the present figures has been further improved in the revised manuscript.

Response to Reviewer 2

Comments and suggestions from Reviewer 2.

The authors have addressed my comments.

Response: Thank you very much for the kind comment.